# Fatty acid remodeling by LPCAT3 enriches arachidonate in phospholipid membranes and regulates triglyceride transport

Tomomi Hashidate-Yoshida[1†], Takeshi Harayama[1†], Daisuke Hishikawa[1], Ryo Morimoto[1,2], Fumie Hamano[2,3], Suzumi M Tokuoka[2], Miki Eto[1,2], Miwa Tamura-Nakano[4], Rieko Yanobu-Takanashi[5], Yoshiko Mukumoto[6], Hiroshi Kiyonari[6], Tadashi Okamura[5,7], Yoshihiro Kita[2,3], Hideo Shindou[1,8], Takao Shimizu[1,2]*

[1]Department of Lipid Signaling, National Center for Global Health and Medicine, Tokyo, Japan; [2]Department of Lipidomics, Graduate School of Medicine, The University of Tokyo, Tokyo, Japan; [3]Life Sciences Core Facility, Graduate School of Medicine, The University of Tokyo, Tokyo, Japan; [4]Communal Laboratory, National Center for Global Health and Medicine, Tokyo, Japan; [5]Department of Laboratory Animal Medicine, National Center for Global Health and Medicine, Tokyo, Japan; [6]Laboratory for Animal Resources and Genetic Engineering, RIKEN Center for Developmental Biology, Kobe, Japan; [7]Section of Animal Models, Department of Infectious Diseases, National Center for Global Health and Medicine, Tokyo, Japan; [8]Core Research for Evolutionary Science and Technology, Japan Science and Technology Agency, Kawaguchi, Japan

*For correspondence: tshimizu@ri.ncgm.go.jp

†These authors contributed equally to this work

**Abstract** Polyunsaturated fatty acids (PUFAs) in phospholipids affect the physical properties of membranes, but it is unclear which biological processes are influenced by their regulation. For example, the functions of membrane arachidonate that are independent of a precursor role for eicosanoid synthesis remain largely unknown. Here, we show that the lack of lysophosphatidylcholine acyltransferase 3 (LPCAT3) leads to drastic reductions in membrane arachidonate levels, and that LPCAT3-deficient mice are neonatally lethal due to an extensive triacylglycerol (TG) accumulation and dysfunction in enterocytes. We found that high levels of PUFAs in membranes enable TGs to locally cluster in high density, and that this clustering promotes efficient TG transfer. We propose a model of local arachidonate enrichment by LPCAT3 to generate a distinct pool of TG in membranes, which is required for normal directionality of TG transfer and lipoprotein assembly in the liver and enterocytes.

## Introduction

Polyunsaturated fatty acids (PUFAs) have important biological functions in health and disease. They are related to cardiovascular diseases, neuronal functions, metabolic syndromes, and many other pathophysiological events (*Poudyal et al., 2011*; *Baum et al., 2012*; *Jiao et al., 2014*). Phospholipids containing PUFAs render the membrane more flexible, which reduces the energy required for deformation (*Rawicz et al., 2000*; *Pinot et al., 2014*). Although PUFAs in phospholipids affect membrane physical properties, the biological significance related to these different properties remain unclear. A recent elegant study showed that PUFAs in phospholipids facilitate endocytosis by reducing the energy required for membrane deformation and fission (*Pinot et al., 2014*). However, the requirement for PUFAs in other biological processes accompanying membrane deformation has to be demonstrated.

**eLife digest** Membranes made of molecules called lipids surround every living cell and also form compartments inside the cell. There are hundreds of different lipid molecules that can be found in membranes. The amount of each type within the membrane can vary, which affects the flexibility and other physical properties of the membrane.

One type of lipid found in membranes is called arachidonic acid. It is involved in cell communication and other processes, and is required for young animals to grow and develop properly. An enzyme called LPCAT3 is thought to incorporate arachidonic acid into membranes, but this has not yet been proven to occur in living animals.

Here, Hashidate-Yoshida, Harayama et al. studied the role of LPCAT3 in newborn mice. The experiments show that this enzyme is found at high levels in the intestine and liver. Mice that lacked LPCAT3 had much lower levels of arachidonic acid compared with normal mice. These mice also showed signs of severe intestinal damage due to the build up of lipids from their mother's milk, and died within a few days of being born.

The mice that lacked LPCAT3 had different amounts of another type of lipid—called triacylglycerols—in their intestine and liver. Normally, these lipids would be assembled into larger molecules called lipoproteins that are released into the blood stream and used in the muscles and other parts of the body. However, Hashidate-Yoshida, Harayama et al. found that in the mice missing LPCAT3, the triacylglycerols did not get assembled into lipoproteins and so they accumulated inside the intestine and liver cells.

The experiments also show that high levels of arachidonic acid and other similar lipids in the membrane enable triacylglycerol molecules to cluster together, which increases the production of lipoproteins. Hashidate-Yoshida, Harayama et al.'s findings suggest that LPCAT3 incorporates arachidonic acid into the membrane of intestine and liver cells, which enables triacylglycerols to be assembled into lipoproteins. The next challenge will be to find out if LPCAT3 is also important for the production of lipoproteins in humans. If it is, then developing new therapies that alter the activity of this enzyme might be beneficial for patients with abnormal levels of lipids in the blood (known as dyslipidemia).

Arachidonate (C20:4; indicative of a fatty acid of 20 carbons with four double bonds) is one of the most studied PUFAs, since it serves as a precursor for biologically active eicosanoids (prostaglandins, leukotrienes, etc) (*Shimizu, 2009*; *Narumiya and Furuyashiki, 2011*). The functions of membrane arachidonate that are not related to eicosanoid synthesis remain largely unknown. We reported previously that arachidonate in membrane phosphatidylcholine (PC) suppresses Akt signaling in cultured cells (*Koeberle et al., 2013*), but it is unknown whether this occurs in vivo. Studies of mice lacking lysophosphatidylinositol acyltransferase 1 revealed that a decrease in phosphatidylinositol (PI) arachidonate has deleterious effects in neuronal development, probably related to PI poly-phosphate dysfunction (*Lee et al., 2012*; *Anderson et al., 2013*). However, the in vivo functions of arachidonate in the most abundant membrane phospholipids such as PC and phosphatidylethanolamine (PE), as well as its effect on membrane physical properties, remained to be solved.

We recently reported that the levels of several PUFAs (such as linoleate [C18:2] and docosahexaenoate [C22:6]) in PC are affected by the substrate selectivity of lysophosphatidic acid acyltransferases (LPAATs) during de novo biosynthesis (*Harayama et al., 2014*), but arachidonate levels seem to be regulated in a different manner. Pioneering studies suggested that arachidonate is incorporated into phospholipids after de novo synthesis during the remodeling pathway (also termed Lands' cycle), which consists of a phospholipid deacylation–reacylation cycle (*Hill and Lands, 1968*). The reacylation is catalyzed by lysophospholipid acyltransferases (LPLATs) (*Hishikawa et al., 2014*), among which lysophosphatidylcholine acyltransferase 3 (LPCAT3) is the strongest candidate to incorporate arachidonate into membranes, due to its substrate selectivity in vitro (*Gijón et al., 2008*; *Hishikawa et al., 2008*; *Matsuda et al., 2008*; *Zhao et al., 2008*). Indeed, knockdown of LPCAT3 by RNA interference reduced the levels of several arachidonate-containing PC species, but additional PUFA-containing PC species were differently affected between studies (*Hishikawa et al., 2008*; *Li et al., 2012*; *Rong et al., 2013*). Therefore, it remains unclear whether LPCAT3 is specific for the

regulation of arachidonate, or it has a broader impact on membrane PUFA levels. In addition, the biological functions of LPCAT3 and membrane arachidonate remain to be fully established, since residual activity of LPCAT3 might affect the overall results in RNA interference experiments.

Since PUFAs affect the bending rigidity of the membrane, it is possible that the regulation of phospholipid arachidonate has effects on membrane shape change. When triglycerides (TGs) are synthesized by diacylglycerol acyltransferases, they accumulate between the two leaflets of the lipid bilayer and deform the membrane (*Yen et al., 2008*; *Thiam et al., 2013*). Molecular dynamics simulation showed that TGs between leaflets generate a 'blister-like' shape, where the membrane on its surface is highly curved (*Khandelia et al., 2010*). Therefore, the ability of the membrane to form curved structures might affect the properties of the TG pool between the leaflets. This pool serves as a precursor for cytosolic lipid droplets in most cells (*Thiam et al., 2013*). In hepatocytes and enterocytes, this pool is also used for transport by microsomal triglyceride transfer protein (MTP) into the endoplasmic reticulum (ER) lumen to generate lipoproteins and luminal lipid droplets (*Sturley and Hussain, 2012*). MTP transfers TGs to nascent apolipoprotein B (apoB) to form primordial lipoproteins (*Abumrad and Davidson, 2012*). MTP also transfers TGs to generate apoB-free lipid droplets in the ER lumen (*Kulinski et al., 2002*), which are required for TG enrichment in nascent lipoproteins. MTP deficiency in humans leads to an impaired absorption of dietary lipids, which is called abetalipoproteinemia (*Wetterau et al., 1992*). Therefore, the normal luminal transfer of TG is critical in vivo, but factor(s) and environment enabling an efficient transport by MTP remain poorly understood (*Yao et al., 2013*).

Here, using LPCAT3-deficient cells and mice, we report that LPCAT3 is critical and relatively specific for the regulation of arachidonate levels in membrane phospholipids in vivo. In addition, we show that LPCAT3 regulates the directionality of TG transfer into lipoproteins, preventing the over-accumulation of cytosolic lipid droplets in hepatocytes and enterocytes. Additional analyses suggest that membrane PUFAs facilitate TG local clustering, which enables efficient transport by MTP. This study identifies LPCAT3 as a major regulator of membrane phospholipid arachidonate, which is a critical factor affecting luminal TG transport.

## Results

### LPCAT3 selectively incorporates arachidonate in membrane phospholipids

To investigate whether the remodeling pathway contributes to arachidonate incorporation into phospholipids, we analyzed the enzymatic properties of LPCAT3. We first examined whether LPCAT3 regulates the acyl-chain composition of the major phospholipid PC. We established rat hepatoma RH 7777 cells that stably overexpress FLAG-tagged murine LPCAT3 (*Figure 1—figure supplement 1A*). Control cells had an LPCAT activity selective for linoleoyl- and arachidonoyl-CoA (*Figure 1A,B*). The overexpression of LPCAT3 doubled LPCAT activity without affecting acyl-CoA selectivity (*Figure 1A,B*). The acyl-chain composition of PC in LPCAT3-overexpressing cells was analyzed using liquid chromatography-tandem mass spectrometry (LC-MS). Chromatograms of 38:4 PC (PC species containing 38 carbons and four double bonds as a sum of the two acyl-chains) revealed two isomers with different retention times (*Figure 1—figure supplement 2*). The levels of 36:4 PC and the later-eluting isomer of 38:4 PC, but not 34:2 PC, were increased in LPCAT3-overexpressing cells (*Figure 1C*). Additional selected reaction monitoring (SRM, see 'Materials and methods' for detail) analyses were performed to resolve the two acyl-chains, and showed that 34:2 PC, 36:4 PC, and the later-eluting isomer (peak 2) of 38:4 PC correspond to 16:0–18:2 PC, 16:0–20:4 PC, and 18:0–20:4 PC, respectively (*Figure 1—figure supplement 2* and data not shown). Therefore, LPCAT3 over-expression increases arachidonate, but not linoleate levels in PC.

Next, LPCAT3-null RH 7777 cells were generated using the CRISPR/Cas9 (Clustered Regularly Interspaced Short Palindromic Repeat/Cas9 [*Ran et al., 2013*]) system for loss of function studies (*Figure 1—figure supplement 3A,B*), and two clones per group were analyzed. LPCAT3-null cells had almost no lysophosphatidylcholine acyltransferase (LPCAT) activity (*Figure 1D*). LPCAT3 deficiency largely decreased arachidonate-containing 36:4 PC and 38:4 PC levels, but not linoleate-containing 34:2 PC (*Figure 1E*). The changes in LPCAT activity and levels of arachidonate-containing PC were restored when deficient cells were stably transfected with wild type murine LPCAT3, but not with a mutant H374A (*Shindou et al., 2009*) lacking enzymatic activity (*Figure 1F,G*, and *Figure 1—figure*

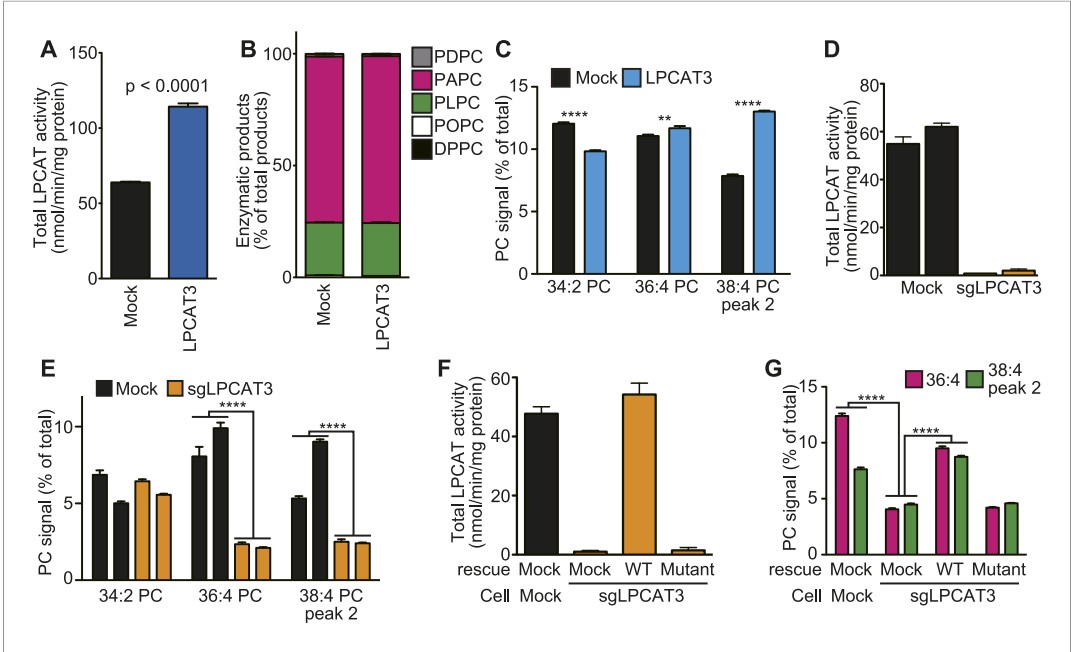

**Figure 1**. LPCAT3 regulates arachidonate levels in PC. (**A** and **B**) LPCAT activity and selectivity in mock- or LPCAT3-overexpressing RH 7777 cells. LPC and five acyl-CoA species (16:0-, 18:1-, 18:2-, 20:4-, and 22:6-CoA) were used as substrates of LPCAT assay to generate DPPC (16:0–16:0 PC), POPC (16:0–18:1 PC), PLPC (16:0–18:2 PC), PAPC (16:0–20:4 PC), and PDPC (16:0–22:6 PC). The total of five products (**A**) and the relative amount of each PC species, equivalent to acyl-CoA selectivity (**B**) are shown. The experiment was performed in technical triplicate. (**C**) Levels of selected PC species in mock- or LPCAT3-stable cells (n = 8). Data are % of the total signals from all PC species detected. (**D–G**) Total LPCAT activity products (**D** and **F**, in technical triplicate) and PC levels of the selected species (**E** (n = 3) and **G** (n = 5)) were measured in mock- or LPCAT3-null RH 7777 cells (**D** and **E**) or the null cells rescued with LPCAT3 or a mutant lacking activity (**F** and **G**). (**D** and **E**) The results of two different clones are separately indicated. Error bars are SD (**A**, **B**, **D**, and **F**) or SEM (**C**, **E**, and **G**). sgLPCAT3: single guide RNA targeting *Lpcat3*. **p < 0.01, ****p < 0.0001. See also *Figure 1—figure supplements 1–3*.

The following figure supplements are available for figure 1:

**Figure supplement 1**. Confirmation of stable expression.

**Figure supplement 2**. Annotation of LC-MS signals.

**Figure supplement 3**. Generation of LPCAT3-null cells.

supplement 1B). Taken together with overexpression studies, despite its enzymatic preference for both linoleate and arachidonate in vitro, LPCAT3 has a high selectivity on the accumulation of arachidonate-containing PC at the cellular level.

## LPCAT3 is the major LPCAT enzyme in vivo

Since the cell culture experiments established LPCAT3 as a target molecule to investigate the functions of arachidonate in membrane phospholipids, we next analyzed the functions of this enzyme in vivo. We first examined the expression pattern of LPCAT3 at the late embryonic stages (between embryonic days 18.5 (E18.5) and E19.5). LPCAT3 mRNA was ubiquitously detected, with high expression in intestine, followed by liver (*Figure 2A*), consistent with the protein level (*Figure 2B*). Intestinal LPCAT3 mRNA expression was strongly induced during the late developmental stages, while the changes in the liver were relatively small (*Figure 2C*). To examine intestinal LPCAT3 expression more in detail, we divided the whole intestine into three parts: the proximal small intestine, where lipid absorption is high (*Abumrad and Davidson, 2012*), distal small intestine, and colon

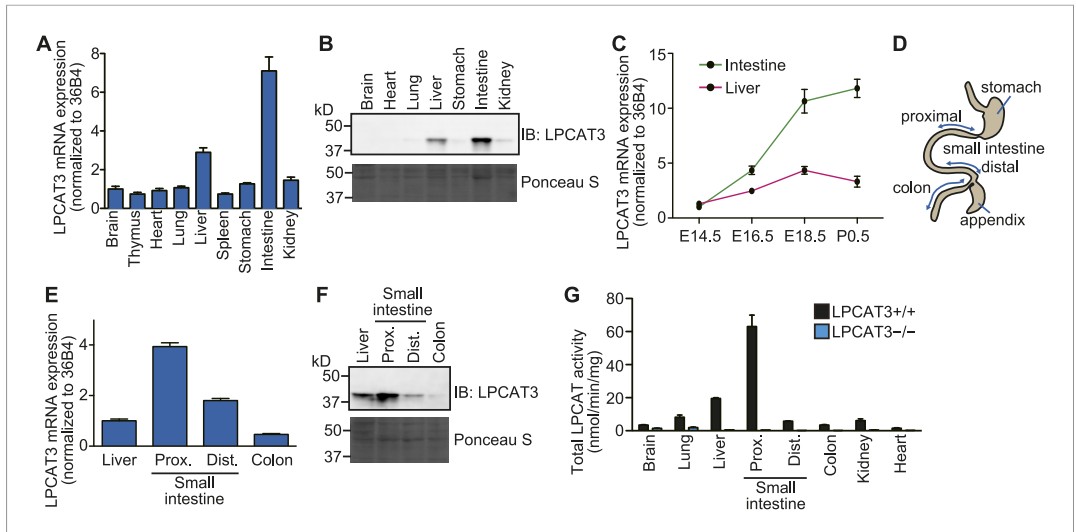

**Figure 2**. Tissue distribution of LPCAT3. (**A**–**F**) The distribution of LPCAT3 mRNA (**A**, **C**, and **E**, by quantitative PCR) or protein (**B** and **F**, by western blot analysis) was analyzed in different tissues at E18.5 (**A**, **B**, **E**, and **F**) or at various perinatal stages (**C**). Ponceau S staining was performed as a loading control for western blot analysis. (**D**) Illustration of the intestine separated into the proximal-, distal small intestine, and colon. (**G**) LPCAT activity was determined in tissues obtained from mice of the indicated genotype at E18.5. See legend of *Figure 1A* for description of the method. Error bars are SEM (n = 3). See also *Figure 2—figure supplement 1*.

The following figure supplement is available for figure 2:

**Figure supplement 1**. Generation of LPCAT3-deficient mice.

(*Figure 2D*). LPCAT3 expression was highest in the proximal small intestine (*Figure 2E,F*). Enzymatic assays revealed that LPCAT activity is highly correlated with LPCAT3 expression (*Figure 2B,F,G*). We generated LPCAT3-deficient mice by gene targeting and confirmed the loss of protein expression (*Figure 2—figure supplement 1A–C*). LPCAT activity was almost completely absent in tissues of LPCAT3-deficient mice (*Figure 2G*). Therefore, LPCAT3 is the major LPCAT enzyme in vivo.

## LPCAT3 regulates arachidonate-containing phospholipids in vivo

We next analyzed how LPCAT3 deficiency affects the lipidome in mice at E18.5–E19.5. We measured tissue PC levels and found a slight reduction in the proximal small intestine of LPCAT3-deficient mice, but not in other tissues (*Figure 3A*). Next, we analyzed the changes in acyl-chain composition of phospholipids from tissues of LPCAT3-deficient mice. LC-SRM-MS lipidomics analysis of four phospholipid classes (PC, PE, phosphatidylserine (PS), and PI) was carried out. The total of signals from all the detected molecules were comparable between genotypes, and were used for normalization (*Figure 3—figure supplement 1*). The slight reduction in the proximal small intestine of LPCAT3-deficient mice was consistent with the reduced PC amount (*Figure 3A* and *Figure 3—figure supplement 1*). To detect the global differences in PC, we analyzed the ratio (as % of wild type) for each PC species in individual tissues, and then averaged the ratio values from all tissues (*Figure 3B*). This analysis enabled us to detect the most common changes that occur under LPCAT3 deficiency. The most decreased species were 34:4 PC, 36:4 PC, 36:5 PC, and 38:4 PC (shown in green in *Figure 3B*), suggesting that the reduction in arachidonate levels is most prominent. These decreases occurred in almost all tissues analyzed (*Figure 3C–E*). The changes in species that might contain linoleate (34:2 PC and 36:2 PC) were small or absent (*Figure 3B*). On the other hand, the most increased PC species were 40:4 PC, 40:5 PC, 42:4 PC, and 42:5 PC (shown in magenta in *Figure 3B*), which might contain fatty acids that arise from arachidonate metabolism (carbon chain elongation, desaturation, and partial β-oxidation [*Schmitz and Ecker, 2008*] [*Figure 3B,F,G*]). To investigate the acyl-chain composition of the changed PC species (34:4 PC, 36:4 PC, 36:5 PC, 38:4 PC, 40:4 PC, 40:5 PC, 42:4 PC, and 42:5 PC), we performed SRM analyses of PC from the proximal small intestine, using

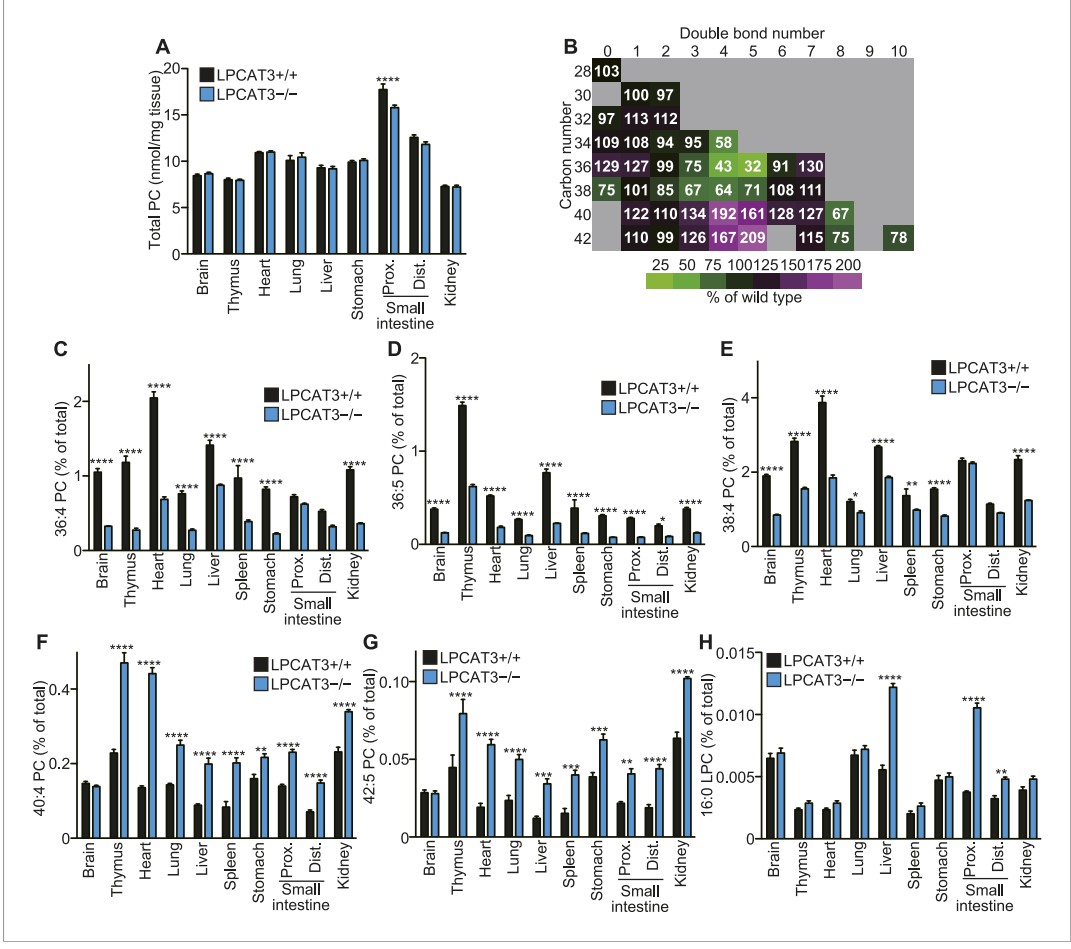

**Figure 3**. LPCAT3 regulates PC arachidonate levels in vivo. Lipidomics analyses were performed in various tissues of mice at E18.5. (**A**) Total PC amounts in tissues from wild type and LPCAT3-deficient mice. (**B**) Heat map showing the ratio of each PC species in LPCAT3-deficient mice. Each value is the average of ratio values (% of wild type) for tissues analyzed in (**C–H**). (**C–G**) The levels of the indicated PC species in wild type and LPCAT3-deficient mice were measured by LC-MS. (**H**) The levels of 16:0 LPC were measured in wild type and LPCAT3-deficient mice. Data were normalized to total MS signals shown in *Figure 3—figure supplement 1*. Prox.: Proximal; Dist.: Distal. (**A**, and **C–H**) Error bars are SEM (n = 5). *p < 0.05, **p < 0.01, ***p < 0.001, ****p < 0.0001. See also *Figure 3—figure supplements 1, 2*.

The following figure supplements are available for figure 3:

**Figure supplement 1**. Total signals used for normalization of lipidomic profiling analysis.

**Figure supplement 2**. Metabolism of arachidonate.

transitions that discriminate fatty acids (*Table 1*). Each PC subspecies did not have a single acyl-chain composition, but was a mixture of various combinations. Interestingly, most of arachidonate-containing species were decreased in LPCAT3-deficient mice, as well as some species containing C18 PUFAs (C18:2 and C18:3) and other C20 PUFAs (C20:3 and C20:5). On the other hand, PC species containing C22 PUFAs (C22:4 and C22:5) and C24 PUFAs (C24:4 and C24:5) were increased, suggesting that the excess of arachidonoyl-CoA caused by LPCAT3 deficiency increases its metabolism (such as elongation) into C22 and C24 PUFAs, which were then incorporated into PC (*Figure 3—figure supplement 2*) (*Schmitz and Ecker, 2008*). In addition to the changes in PC, we found that lysophosphatidylcholine (LPC) levels are increased in the proximal small intestine, distal small intestine, and liver of LPCAT3-deficient mice (*Figure 3H*).

**Table 1**. Acyl-chain composition of PC species with changed levels in LPCAT3-deficient mice

| PC subclass | Acyl-chain composition | LPCAT3+/+ peak area | | LPCAT3−/− peak area | | KO/WT (%) | p value |
|---|---|---|---|---|---|---|---|
| | | Mean | SEM | Mean | SEM | | |
| 34:4 | 14:0–20:4 | 32,607 | 1253 | 11,609 | 835 | 36 | <0.0001 |
| | 16:1–18:3 | 42,145 | 2574 | 41,340 | 4555 | 98 | 0.8816 |
| 36:4 | 16:0–20:4 | 1,037,887 | 21,426 | 917,642 | 8844 | 88 | 0.0008 |
| | 18:1–18:3 | 122,575 | 8421 | 61,410 | 1726 | 50 | 0.0001 |
| | 18:2–18:2 | 366,675 | 17,961 | 250,350 | 26,185 | 68 | 0.0064 |
| 36:5 | 16:0–20:5 | 147,054 | 8925 | 3872 | 1095 | 3 | <0.0001 |
| | 16:1–20:4 | 104,768 | 7773 | 79,851 | 4883 | 76 | 0.0265 |
| | 18:2–18:3 | 36,381 | 3669 | 16,762 | 1790 | 46 | 0.0013 |
| 38:4 | 16:0–22:4 | 81,093 | 6150 | 169,849 | 9408 | 209 | <0.0001 |
| | 18:0–20:4 | 670,488 | 23,121 | 624,853 | 18,289 | 93 | 0.1602 |
| | 18:1–20:3 | 230,517 | 10,302 | 108,788 | 8009 | 47 | <0.0001 |
| | 18:2–20:2 | 29,789 | 1587 | 14,567 | 701 | 49 | <0.0001 |
| 40:4 | 16:0–24:4 | 10,392 | 1617 | 20,865 | 3072 | 201 | 0.0167 |
| | 18:0–22:4 | 176,325 | 4788 | 293,832 | 6785 | 167 | <0.0001 |
| | 18:1–22:3 | 22,450 | 2704 | 62,145 | 3805 | 277 | <0.0001 |
| | 20:0–20:4 | 12,131 | 1265 | 6590 | 899 | 54 | 0.0073 |
| | 20:1–20:3 | 32,191 | 2591 | 4991 | 1740 | 16 | <0.0001 |
| 40:5 | 16:0–24:5 | 1535 | 1258 | 5381 | 1162 | 350 | 0.0549 |
| | 18:0–22:5 | 110,752 | 7361 | 163,723 | 14,186 | 148 | 0.0106 |
| | 18:1–22:4 | 39,603 | 2862 | 117,362 | 5425 | 296 | <0.0001 |
| | 20:1–20:4 | 58,465 | 3767 | 32,595 | 2075 | 56 | 0.0003 |
| 42:4 | 18:0–24:4 | 1907 | 403 | 12,412 | 1762 | 651 | 0.0004 |
| | 18:1–24:3 | 1663 | 777 | 9584 | 639 | 576 | <0.0001 |
| | 18:2–24:2 | 4497 | 950 | 1587 | 605 | 35 | 0.0736 |
| | 22:0–20:4 | 11,180 | 930 | 5039 | 1154 | 45 | 0.0032 |
| 42:5 | 18:0–24:5 | 3092 | 629 | 14,907 | 882 | 482 | <0.0001 |
| | 18:1–24:4 | 1258 | 405 | 11,476 | 858 | 912 | <0.0001 |
| | 22:1–20:4 | 25,450 | 2790 | 11,728 | 1853 | 46 | 0.0035 |

SRM analyses for discrimination of fatty acids were performed for the indicated PC species (n = 5). Anions of the fatty acids indicated at the second position (e.g., 20:4 for 14:0–20:4 PC) were used for selection at Q3. Sample concentration was adjusted based on tissue weight.

Analysis of other phospholipid classes showed that 38:4 PE, but not 38:4 PI levels are decreased, and that 40:4 PE and 40:5 PE are increased in LPCAT3-deficient mice (*Figure 4A–D*). Similarly with PC, SRM analyses for fatty acid discrimination revealed that PE species containing C20 PUFAs are decreased, while those containing C22 PUFAs are increased in LPCAT3-deficient mice (*Table 2*). This is consistent with the LPEAT activity of LPCAT3 with arachidonate preference (*Figure 4E,F*). The number of detected molecules for PS was low, thus we did not analyze these species in detail. However, preliminary data suggested that arachidonate-containing PS species are decreased in LPCAT3-deficient mice (data not shown).

To gain further insights in the fatty acid compositions, we performed gas chromatography with flame ionization detection (GC-FID). Fatty acids were detected as methyl esters derivatized from total lipids (including free fatty acids, neutral lipids, and phospholipids). The total amounts of fatty acids were slightly decreased in proximal small intestine of LPCAT3-deficient mice, as expected from the reduced PC amount (*Figure 3A* and *Figure 5—figure supplement 1*). Unexpectedly, they increased

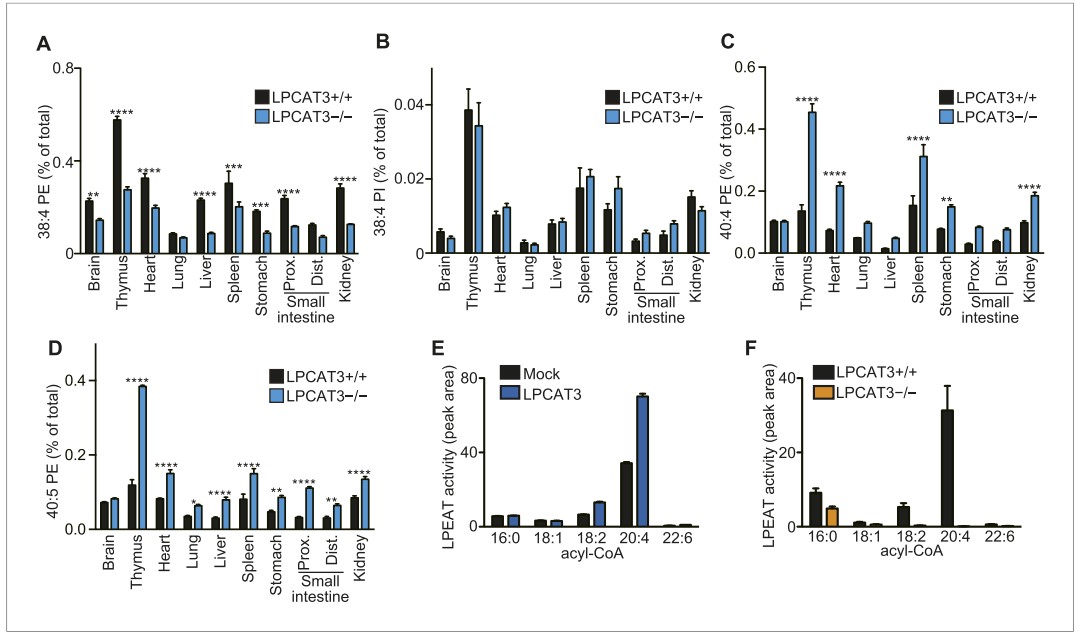

**Figure 4.** LPCAT3 regulates PE arachidonate levels in vivo. (**A–D**) The levels of the indicated PE species in wild type and LPCAT3-deficient mice were measured by LC-MS and normalized using the values in *Figure 3—figure supplement 1*. Error bars are SEM (n = 5). (**E** and **F**) LPEAT activity was examined in mock- or LPCAT3-transfected RH 7777 cells (error bars are SD, technical triplicate) (**E**) and proximal small intestine from wild type or LPCAT3-deficient mice (error bars are SEM, n = 3) (**F**). 17:1 LPE was used instead of LPC in a similar experiment to *Figure 1A*, and peak areas of each LPEAT product are illustrated. Prox.: Proximal; Dist.: Distal. *p < 0.05, **p < 0.01, ***p < 0.001, ****p < 0.0001.

in liver of LPCAT3-deficient mice (*Figure 5—figure supplement 1*). GC-FID confirmed that arachidonate levels are reduced in all tissues of LPCAT3-deficient mice (*Figure 5A*). Linoleate levels were decreased only in some tissues (*Figure 5B*), showing that the changes seen in proximal small intestine PC (*Table 1*) are not universal, consistently with *Figure 3B*. We observed increased docosahexaenoate levels, but only in some tissues (*Figure 5C*). Instead, adrenate (C22:4 n-6) and

**Table 2**. Acyl-chain composition of PE species with changed levels in LPCAT3-deficient mice

| PE subclass | Acyl-chain composition | LPCAT3+/+ peak area | | LPCAT3−/− peak area | | KO/WT (%) | p value |
|---|---|---|---|---|---|---|---|
| | | Mean | SEM | Mean | SEM | | |
| 38:4 | 16:0–22:4 | 373,458 | 20,239 | 1,073,902 | 50,565 | 288 | <0.0001 |
| | 18:0–20:4 | 8,394,810 | 381,499 | 4,349,617 | 102,271 | 52 | <0.0001 |
| | 18:1–20:3 | 1,503,041 | 38,884 | 689,131 | 14,846 | 46 | 0.0002 |
| 40:4 | 18:0–22:4 | 1,831,032 | 80,319 | 3,803,421 | 98,424 | 208 | <0.0001 |
| | 18:1–22:3 | 124,638 | 4846 | 236,224 | 23,357 | 190 | 0.0016 |
| | 20:0–20:4 | 187,358 | 8687 | 45,797 | 4286 | 24 | <0.0001 |
| | 20:1–20:3 | 35,837 | 3019 | 11,721 | 2388 | 33 | 0.0002 |
| 40:5 | 18:0–22:5 | 523,416 | 4657 | 2,056,845 | 111,780 | 393 | <0.0001 |
| | 18:1–22:4 | 731,947 | 43,088 | 1,934,984 | 35,746 | 264 | <0.0001 |
| | 20:1–20:4 | 224,759 | 3572 | 64,708 | 3066 | 29 | <0.0001 |

SRM analyses for discrimination of fatty acids were performed for the indicated PE species (n = 5). Anions of the fatty acids indicated at the second position (e.g., 22:4 for 16:0–22:4 PE) were used for selection at Q3. Sample concentration was adjusted based on tissue weight.

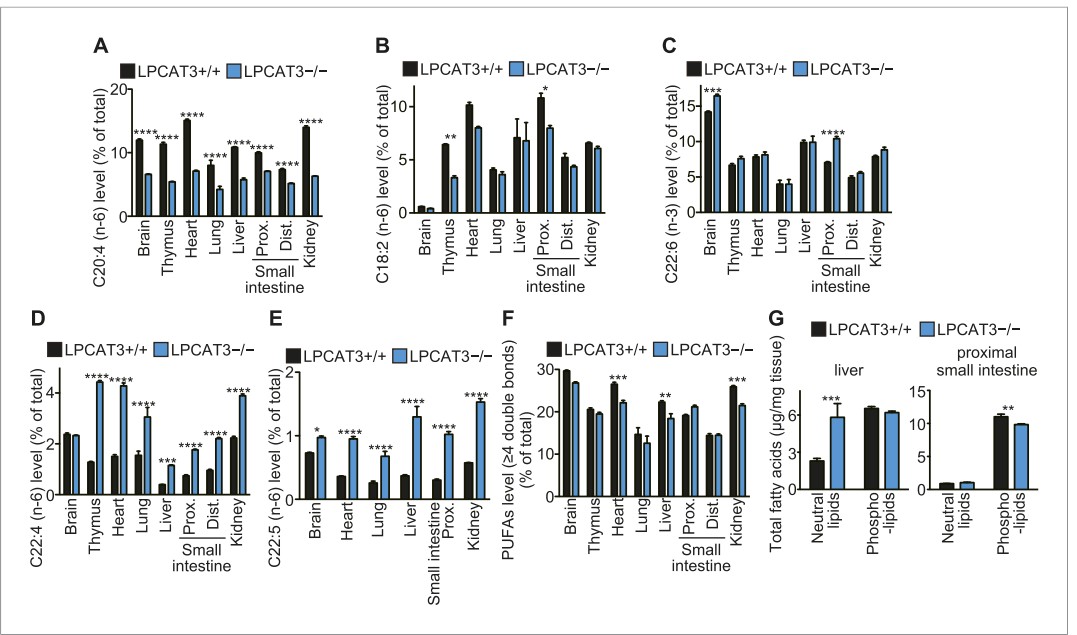

**Figure 5**. Arachidonate levels and metabolism are changed in LPCAT3-deficient mice GC-FID was performed using various tissues to analyze fatty acid amounts and compositions in total lipids, as well as those of lipid subclasses. (**A**–**F**) The levels of the indicated PUFAs (**A**–**E**), and the total of PUFAs (**F**, sum of PUFAs with four or more double bonds) in wild type and LPCAT3-deficient mice were measured by GC-FID. Data are % of the total levels shown in *Figure 5—figure supplement 1*. (**G**) The total amounts of fatty acids in neutral lipids and phospholipids that were fractionated from the indicated tissues in wild type and LPCAT3-deficient mice were analyzed by GC-FID. Prox.: Proximal; Dist.: Distal. Error bars are SEM (n = 5). *p < 0.05, **p < 0.01, ***p < 0.001, ****p < 0.0001. See also *Figure 5—figure supplement 1*.

The following figure supplement is available for figure 5:

**Figure supplement 1**. Total fatty acids in tissues.

docosapentaenoate (C22:5 n-6) levels were increased in almost all tissues, confirming that the excess of arachidonoyl-CoA caused by LPCAT3 deficiency increases its use for further metabolism, as has been suggested by the results of LC-MS (*Figure 5D,E*, and *Figure 3—figure supplement 2*). C24 PUFAs could not be analyzed under the GC-FID condition used. Due to these increases, the total amounts of PUFAs that have four or more double bonds were only slightly affected in LPCAT3-deficient mice (*Figure 5F*).

To investigate the mechanism of the unexpected increased fatty acids in liver from LPCAT3-deficient mice, total liver lipids were divided into neutral lipid and phospholipid fractions by solid phase extraction. We found that fatty acid amounts of neutral lipids are increased in liver of LPCAT3-deficient mice (*Figure 5G*). A similar experiment was performed using proximal small intestine, and fatty acid amounts of phospholipids are decreased in this tissue of LPCAT3-deficient mice (*Figure 5G*), consistent with the decreased PC levels (*Figure 3A*).

Since it was proposed that LPLATs promote the reacylation of free arachidonate and reduce eicosanoid production (*Zarini et al., 2006*; *Gijón et al., 2008*), we investigated eicosanoid levels, but found no major change in LPCAT3-deficient mice, at least under normal conditions (*Figure 6A–C* and data not shown). In conclusion, LPCAT3 regulates membrane phospholipid arachidonate levels without largely affecting the total degree of membrane unsaturation and eicosanoid levels. Therefore, the phenotypes of LPCAT3-deficient mice under normal conditions would reveal the functions of membrane arachidonate that are independent of eicosanoids.

## Accumulation of TGs in LPCAT3-deficient embryonic livers

Next, we analyzed liver of LPCAT3-deficient mice at E18.5–E19.5, because fatty acid levels were increased in neutral lipids, suggesting an accumulation of TGs (*Figure 5G*). GC-FID analysis of the

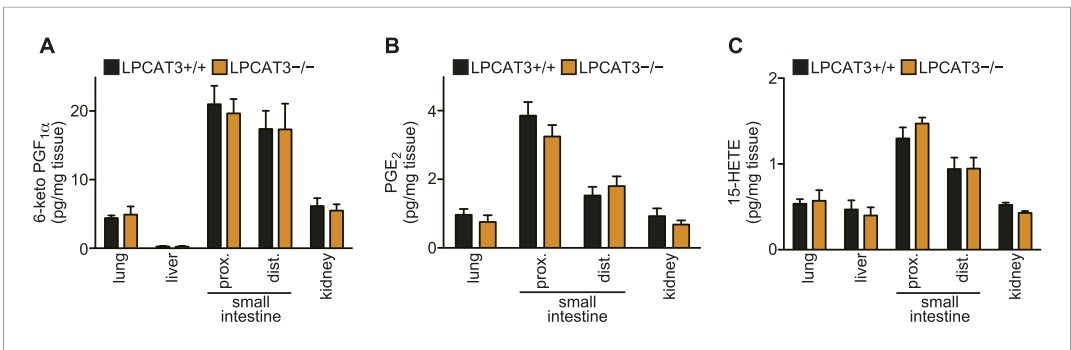

**Figure 6**. Eicosanoid levels are not affected in tissues of LPCAT3-deficient mice. (**A**–**C**) The levels of 6-keto PGF$_1$α (**A**), PGE$_2$ (**B**) and 15-HETE (**C**) in wild type and LPCAT3-deficient mice at E18.5. Error bars are SEM (n = 5). PG: prostaglandin; HETE: hydroxyeicosatetraenoic acid.

phospholipid fraction confirmed that arachidonate levels were decreased in liver of LPCAT3-deficient mice (*Figure 7A*). On the other hand, C22 n-6 PUFAs and docosahexaenoate were increased. Although they were minor components, GC-FID analysis of the neutral lipid fraction showed that arachidonate and C22 n-6 PUFAs (C22:4 and C22:5) are increased in neutral lipids of LPCAT3-deficient mice, while docosahexaenoate was unchanged (*Figure 7B*).

We found that liver TG levels are increased in LPCAT3-deficient mice, while liver cholesterol levels were only slightly changed (*Figure 7C,D*). Histological analysis revealed that a massive amount of glycogen is present at this period (irrespectively of the genotype), and we could not distinguish wild type and LPCAT3-deficient mice by hematoxylin/eosin, or periodic acid-Schiff (PAS)/hematoxylin staining (*Figure 7—figure supplement 1A–D*). On the other hand, oil red O staining revealed an accumulation of lipid droplets in embryonic liver of LPCAT3-deficient mice (*Figure 7E,F*), which was confirmed by electron microscopy (*Figure 7G,H*).

The loss of very low-density lipoprotein (VLDL) assembly might explain such a phenotype, but the protein levels of MTP and protein disulfide isomerase (PDI, an interacting partner of MTP required for TG transfer [*Kulinski et al., 2002*]) appeared normal in LPCAT3-deficient mice (*Figure 7—figure supplement 2A,B*). In addition, VLDL-like particles were detected in the Golgi apparatus of LPCAT3-deficient mice (*Figure 7—figure supplement 2C*). Therefore, LPCAT3-deficient mice accumulate TGs in the liver at embryonic stages, while the components of VLDL assembly are present.

## Lethality and enterocyte damage in LPCAT3-deficient mice

LPCAT3-deficient mice were born at a Mendelian distribution (n = 24 [LPCAT3+/+], 45 [LPCAT3+/−], 21 [LPCAT3−/−]), but died within a few days after birth (*Figure 8A*). Although they looked indistinguishable from control mice at birth and had normal suckling (*Figure 8B*) and breathing, they did not gain weight and became smaller than wild type mice during the first days (*Figure 8C,D*). LPCAT3-deficient mice had normal blood glucose levels at birth (at postnatal day (P)0.5), but became severely hypoglycemic at P1.5 (*Figure 8E,F*). Hypoglycemia was not due to hyperinsulinemia or an insufficient hepatic glycogenolysis (*Figure 8—figure supplement 1A–D*). Histological analysis of multiple tissues from mice at P1.5 was performed, and the most significant difference between wild type and LPCAT3-deficient mice was seen in the small intestine. The proximal small intestine of LPCAT3-deficient mice had a 'vacuolated' appearance (*Figure 9A,B*). Oil red O staining revealed that neutral lipids are normally transported in wild type mice, while they drastically accumulate in intestinal epithelial cells of LPCAT3-deficient mice (*Figure 9C,D*). Massive lipid droplet accumulation was confirmed by electron microscopy (*Figure 9E–H*). In addition to lipid droplet accumulation, enterocytes of LPCAT3-deficient mice were severely damaged, as judged from the loss of microvilli and deformed mitochondria (*Figure 9G,H*). Therefore, this severe damage probably caused a malabsorption of nutrients leading to hypoglycemia. On the other hand, although the arachidonate content was decreased in phospholipids of the small intestine from LPCAT3-deficient embryos at E18.5–E19.5 (*Figure 9I*), their histological appearance was

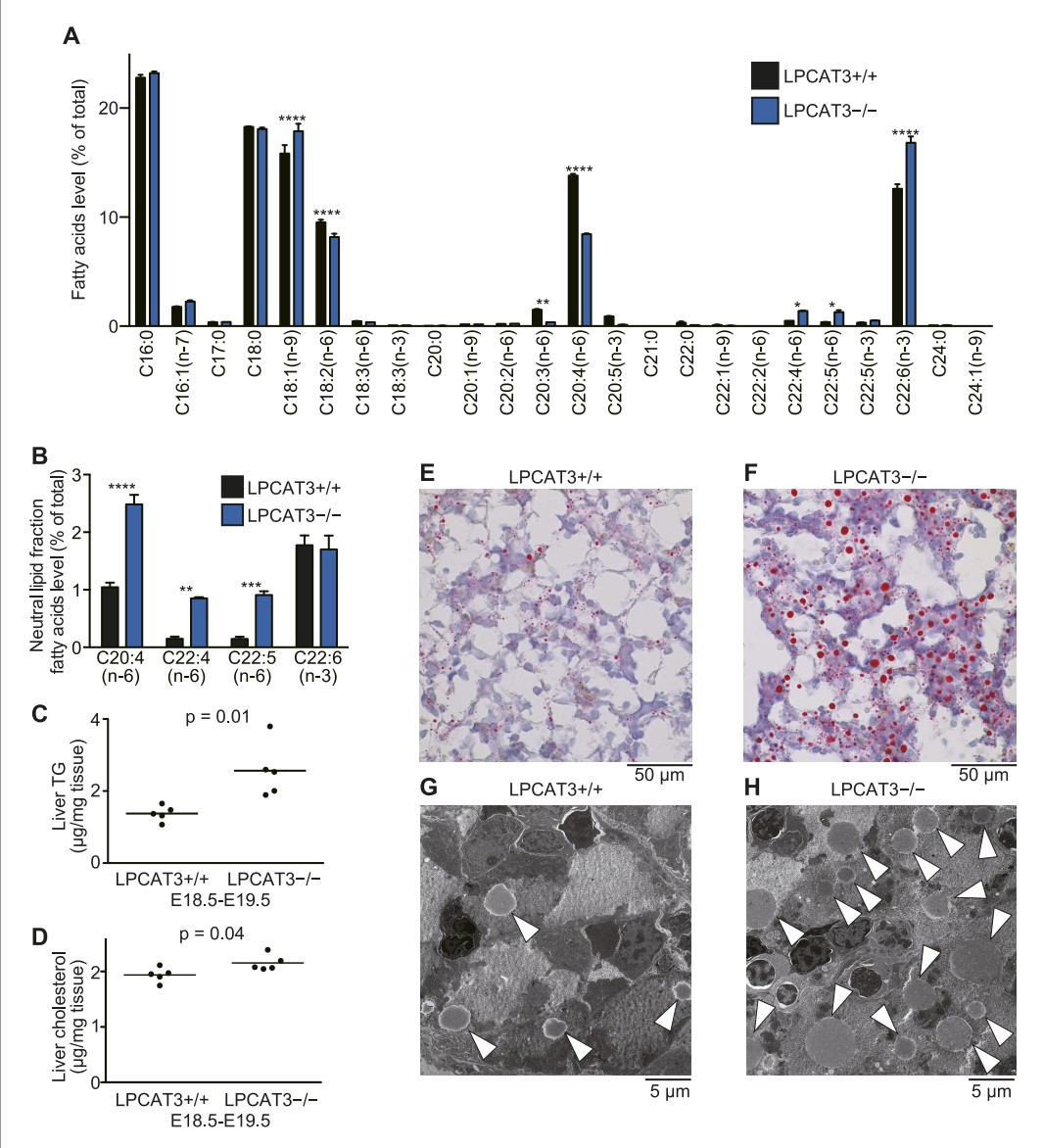

**Figure 7**. TGs accumulate in the embryonic liver of LPCAT3-deficient mice. (**A** and **B**) Liver phospholipids (**A**) and neutral lipids (**B**, only selected PUFAs are shown) were obtained by solid-phase extraction and used for the analysis of acyl-chain composition using wild type and LPCAT3-deficient mice. The percentage of fatty acids detected by GC-FID is illustrated. Error bars are SEM (n = 5). (**C** and **D**) Levels of TG (**C**) and cholesterol (**D**) in liver were measured using wild type and LPCAT3-deficient mice at E18.5–E19.5. (**E**–**H**) The amounts of lipid droplets in liver of wild type and LPCAT3-deficient mice at E18.5–E19.5 were detected by oil red O and hematoxylin staining (**E** and **F**), or by electron miscopy (**G** and **H**). Arrowheads: lipid droplets. *p < 0.05, **p < 0.01, ****p < 0.0001. See also *Figure 7—figure supplements 1, 2*.

The following figure supplements are available for figure 7:

**Figure supplement 1**. Histological analysis of embryonic liver.

**Figure supplement 2**. The machinery for VLDL assembly is present in LPCAT3-deficient liver.

normal (data not shown). Therefore, the intestinal damages in LPCAT3-deficient mice develop postnatally, most likely due to the epithelial accumulation of lipids from breast milk intake soon after birth.

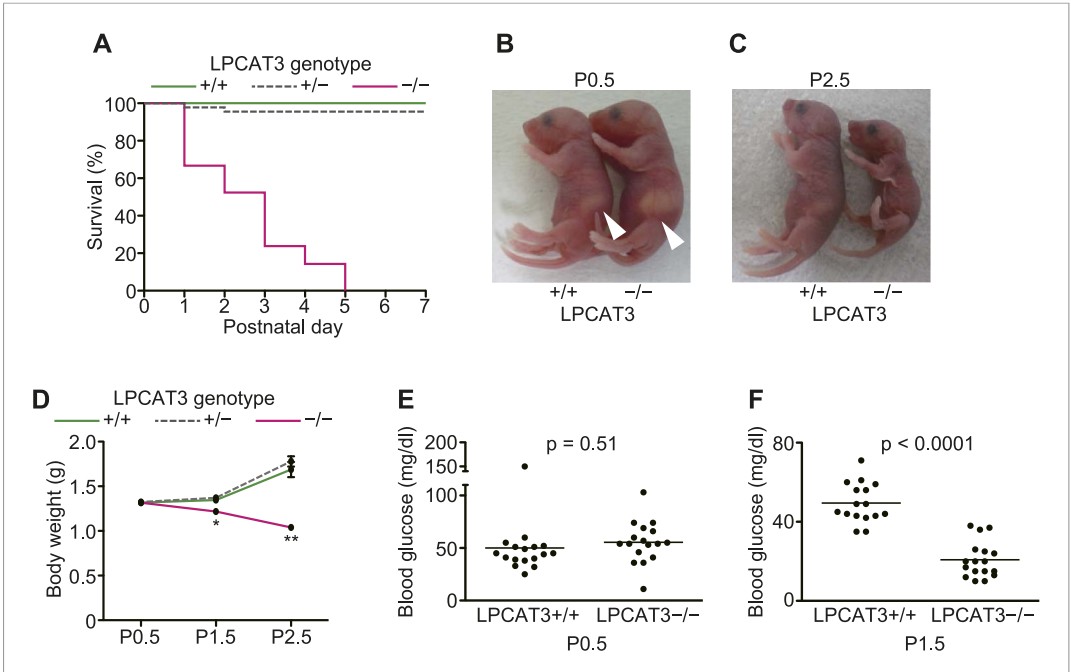

**Figure 8**. LPCAT3-deficient mice are neonatally lethal. (**A**–**D**) The survival rate (**A**), gross appearance (**B** and **C**), and body weight change (n = 9–62) (**D**) were evaluated in neonatal mice of the indicated genotypes. Arrowhead: normal milk intake in both genotypes. (**E** and **F**) Blood glucose was measured in wild type and LPCAT3-deficient mice at P0.5 (**E**) and P1.5 (**F**). Error bars are SEM. *p < 0.05, **p < 0.01. See also *Figure 8—figure supplement 1*.

The following figure supplement is available for figure 8:

**Figure supplement 1**. Plasma insulin and liver glycogen in LPCAT3-deficient mice.

Next, we analyzed tissue neutral lipid levels in small intestine of LPCAT3-deficient mice at P1.5. In the proximal small intestine, TG levels were high due to the massive absorption of lipids at this region, but tended to be increased in LPCAT3-deficient mice (*Figure 9J*). On the other hand, cholesterol did not accumulate in LPCAT3-deficient mice at this region (*Figure 9K*). In addition, we found an increase in TG and cholesterol levels in distal small intestine of LPCAT3-deficient mice at P1.5 (*Figure 9J,K*). Oil red O staining revealed that this region compensatorily absorbs lipids in LPCAT3-deficient mice (*Figure 9—figure supplement 1A,B*), probably due to the inability to absorb them at the proximal small intestine. Similarly to the case of the embryonic liver, the accumulation of lipid droplets was not due to MTP deficiency, since the protein levels of MTP and PDI in proximal small intestine were normal in LPCAT3-deficient mice at E18.5–E19.5 (*Figure 9—figure supplement 2A,B*). Also, we detected chylomicron-like particles in the Golgi apparatus of proximal small intestine enterocytes at P0.5 (*Figure 9—figure supplement 2C*). Although the mRNA level of MTP was decreased in proximal intestine of LPCAT3-deficient mice at P1.5, this was explained as a secondary event by the damage of enterocytes, since villin levels also decreased (*Figure 9—figure supplement 2D,E*). On the other hand, the mRNA level of MTP was increased in distal small intestines of LPCAT3-deficient mice at P1.5, being consistent with the acquisition of an absorptive phenotype as stated above (*Figure 9—figure supplements 1A,B, 2D,E*).

## The assembly of TGs into lipoproteins is limited in LPCAT3-deficient mice

Enterocytes and hepatocytes with lipid droplet accumulation secrete neutral lipids into plasma as apoB-containing lipoproteins (*Abumrad and Davidson, 2012*; *Sturley and Hussain, 2012*). Although lipoproteins were detected by electron microscopy (*Figure 7—figure supplement 2C* and *Figure 9—figure supplement 2C*), their composition might be altered. Therefore, we analyzed plasma lipids at various stages. We found that TG levels, but not apoB, cholesterol, or PC levels, are

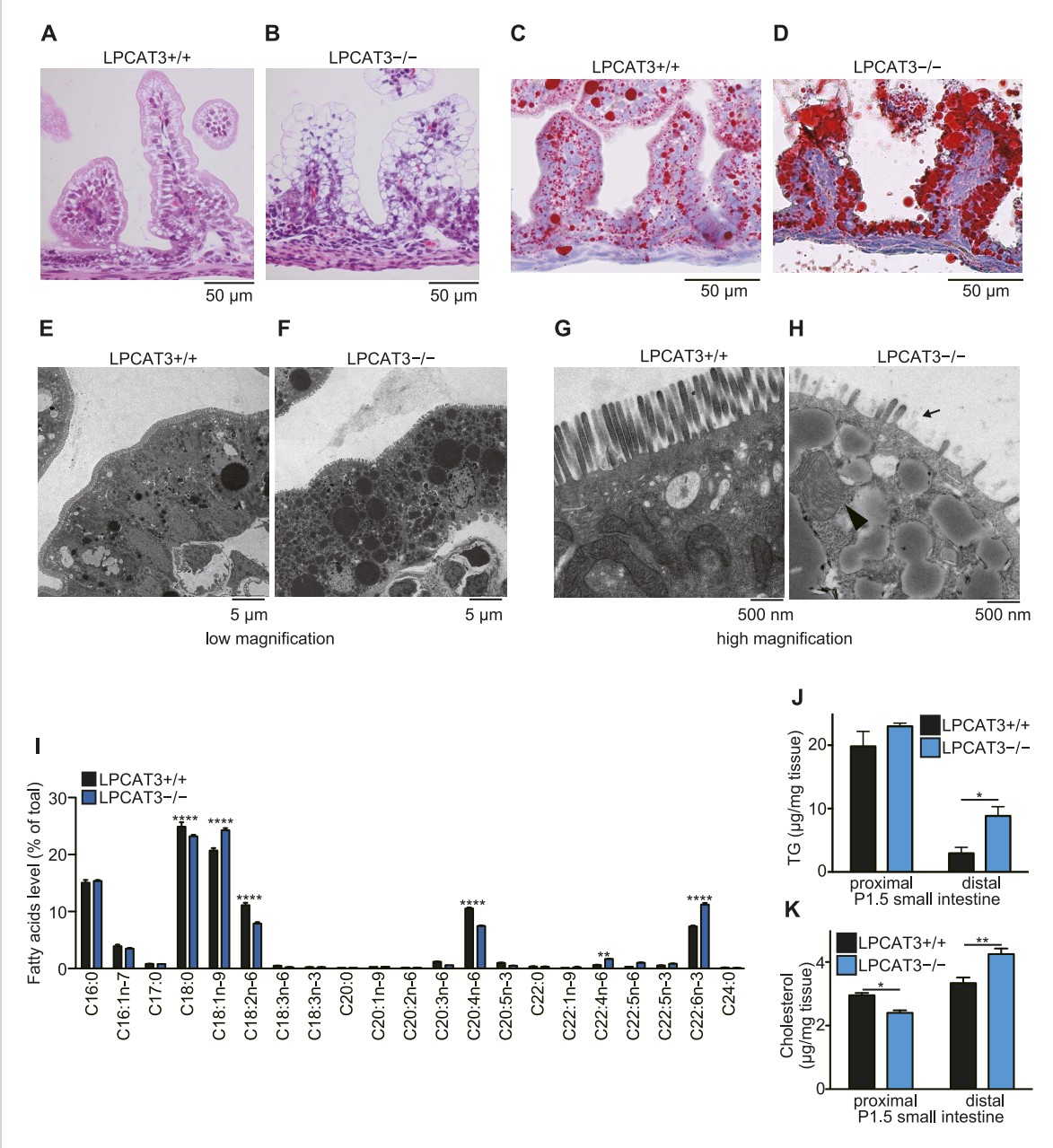

**Figure 9**. TG accumulation in enterocytes of LPCAT3-deficient mice. (**A–H**) Histology of the proximal small intestine of mice at P1.5 was analyzed by light microscopy after staining with hematoxylin and eosin (**A** and **B**), oil red O and hematoxylin (**C** and **D**), or by electron microscopy (**E–H**). (**H**) Shortened microvilli (arrow) and a mitochondrion with disrupted outer membrane (arrowhead) are seen in LPCAT3-deficient mice. (**I**) Phospholipids from proximal small intestine were obtained by solid-phase extraction and used for the analysis of acyl-chain composition using wild type and LPCAT3-deficient mice. The percentage of fatty acids detected by GC-FID is illustrated. (**J** and **K**) Levels of TG (**J**) and cholesterol (**K**) in small intestine samples from wild type and LPCAT3-deficient mice at P1.5. Error bars are SEM (n = 5). *p < 0.05, **p < 0.01. See also *Figure 9—figure supplements 1, 2*.

The following figure supplements are available for figure 9:

**Figure supplement 1**. Distal small intestine of LPCAT3-deficient mice absorbs lipids.

**Figure supplement 2**. The machinery for chylomicron assembly is present in LPCAT3-deficient mice.

significantly decreased in plasma of LPCAT3-deficient mice at E18.5–E19.5 (*Figure 10A–D*). Plasma lipids were further analyzed by gel filtration chromatography. TGs are usually detected in chylomicron or VLDL fractions in adult plasma. However, in our samples, the elution profiles of both TGs and cholesterol suggested an association with particles of an intermediate size between VLDL, intermediate-density lipoprotein (IDL), and low-density lipoprotein (LDL) (*Figure 10—figure supplement 1A–H*). This is consistent with the elution profiles in previous studies (*van Straten et al., 2009*), and thus is a characteristic of this neonatal period. Therefore, we will regard this 'VLDL/IDL/LDL' fraction as the lipoproteins secreted from the liver. TGs in the VLDL/IDL/LDL fraction decreased in LPCAT3-deficient mice at all periods analyzed (*Figure 10E* and *Figure 10—figure supplement 1A–D*). In contrast, cholesterol levels were not decreased (*Figure 10F* and *Figure 10—figure supplement 1E–H*). After P0.4, TGs and cholesterol were detected in the chylomicron fractions (*Figure 10G,H*, and *Figure 10—figure supplement 1A–H*). Chylomicron TG levels were normal in LPCAT3-deficient mice at P0.4, but tended to be decreased at later periods (*Figure 10G* and *Figure 10—figure supplement 1A–D*). Since the chylomicron lipid levels vary largely due to the difficulty to synchronize suckling behaviors, the differences did not reach statistical significance. Chylomicron cholesterol levels were not largely affected in LPCAT3-deficient mice at all periods examined (*Figure 10H* and *Figure 10—figure supplement 1E–H*). Therefore, a reduction in TGs, but not in cholesterol, was observed commonly in lipoproteins secreted from both liver and small intestine. Based on this observation, we calculated the ratio between TGs and cholesterol in chylomicron and VLDL/IDL/LDL fractions at different time points (*Figure 10I,J*). In the chylomicron fraction, the TG/cholesterol ratio increased gradually after birth in wild type, but remained constant after P0.4 in LPCAT3-deficient mice (*Figure 10I*). A similar trend was observed in the VLDL/IDL/LDL fraction (*Figure 10J*). These results suggest that in LPCAT3-deficient mice, the assembly of TG into lipoproteins is normal until some degree, but becomes severely inhibited when high amounts of TGs have to be transported.

To analyze lipoprotein composition more in detail, we used dextran sulfate to precipitate apoB-containing lipoproteins based on their surface charge (*Warnick et al., 1982*). We quantified TG, cholesterol, and PC levels relative to apoB in the precipitates. Consistent with the results from gel filtration chromatography, apoB-associated TG, but not cholesterol or PC, was decreased in LPCAT3-deficient mice (*Figure 10K–M*). The normal PC level was confirmed by gel filtration chromatography (*Figure 10—figure supplement 1I*). Taken together, we conclude that in LPCAT3-deficient mice, TGs are not efficiently used for lipoprotein assembly when high TG amounts have to be transported, leading to cytosolic lipid droplet accumulation and cellular damages.

## Membrane PUFAs enable efficient clustering and transport of TGs

Finally, we investigated how LPCAT3 might affect TG transport. It was unlikely that PC synthesized by LPCAT3 is a major source of lipoprotein PC, since their levels were unchanged in LPCAT3-deficient mice (*Figure 10D,M*, and *Figure 10—figure supplement 1I*). In addition, although the acyl-chain composition of PC in apoB-containing lipoproteins was slightly different in LPCAT3-deficient mice, PC species with arachidonate were relatively minor components of lipoproteins, irrespectively of the genotype (*Figure 11—figure supplement 1*). In addition to these observations, the normal levels of proteins involved in lipoprotein assembly (*Figure 7—figure supplement 2* and *Figure 9—figure supplement 2*) suggested that a previously unknown mechanism should explain the impaired TG transport.

Since the membrane environment around LPCAT3 should be enriched in PUFAs (see 'Discussion'), we analyzed whether a PUFA-rich membrane affects the clustering and transport of TG. When present at a high concentration in a limited space, the fluorescence of the fluorophore 7-nitro-2,1,3-benzoxadiazole (NBD) decreases due to its self-quenching property (*Brown et al., 1994*). Therefore, the quenching of NBD-labeled TG (NBD-TG) in PC liposomes was measured to analyze the local clustering between leaflets (*Figure 11A*). The total fluorescence (without quenching) was measured by disassembling liposomes in isopropanol (*Figure 11A,B*). In the absence of PC, we did not observe NBD-TG fluorescence in the samples, showing that free NBD-TG is lost during liposome preparation (*Figure 11B*). To prepare liposomes, we used egg PC, which contains only a slight amount of PUFAs (*Figure 11—figure supplement 2*), or egg PC supplemented with 30% of synthetic PC containing palmitate at the *sn*-1 position, and oleate, linoleate, arachidonate, or docosahexaenoate at the *sn*-2 position. We calculated the self-quenching of NBD-TG from the fluorescence of liposomes in buffer or isopropanol (*Figure 11B*), and found that a membrane rich in PUFAs such as arachidonate or docosahexaenoate increases quenching (*Figure 11C*), equivalent to local TG clustering.

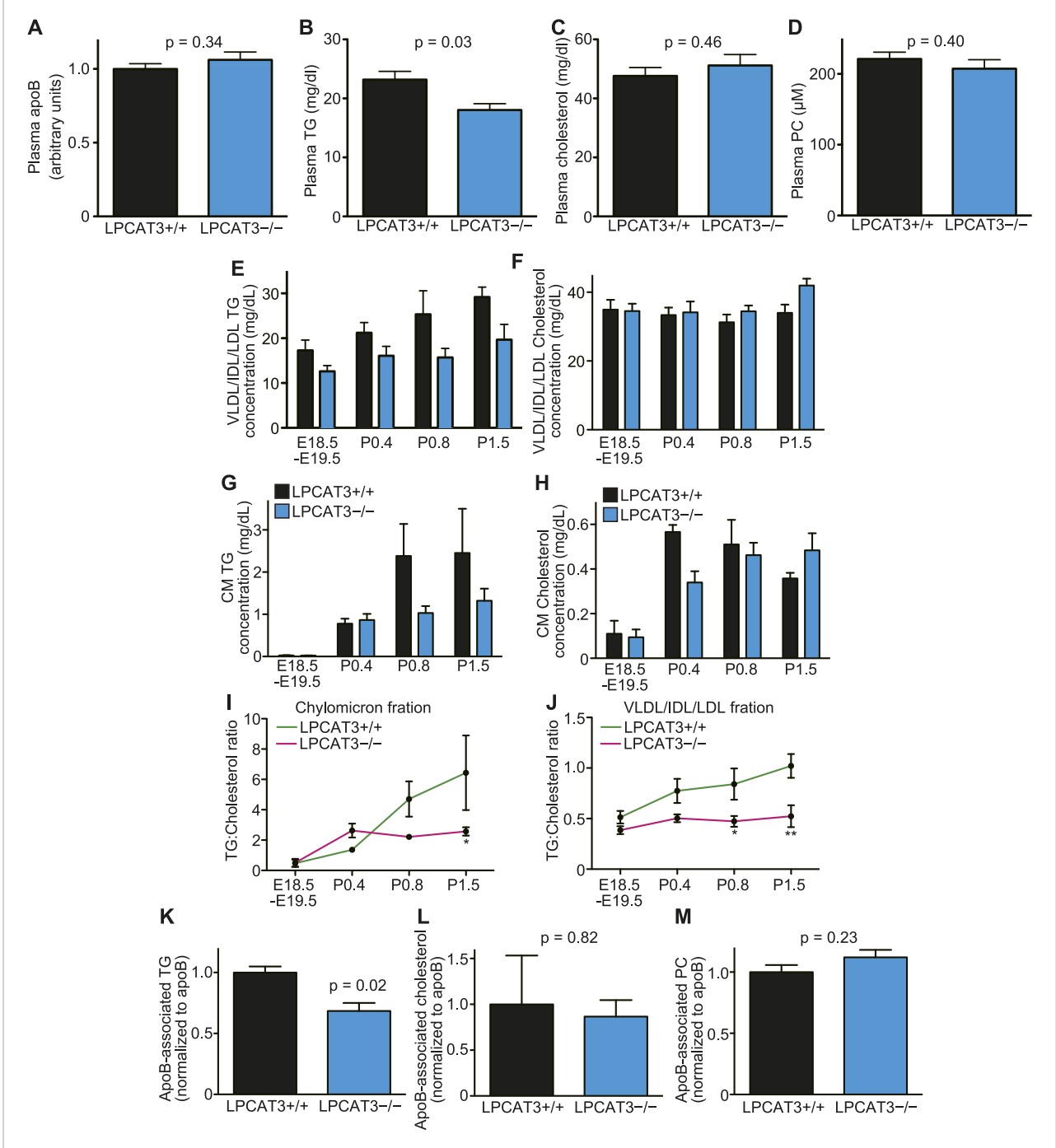

**Figure 10**. The levels of TGs are decreased in plasma lipoproteins of LPCAT3-deficient mice. (**A–D**) The levels of apoB (**A**), TG (**B**), cholesterol (**C**), and PC (**D**) were measured in plasma of wild type and LPCAT3-deficient mice at E18.5–E19.5 (n = 10). (**E–J**) Plasma lipoproteins from wild type and LPCAT3-deficient mice were fractionated by gel-filtration chromatography (n = 5). TG (**E** and **G**) and cholesterol (**F** and **H**) levels in VLDL/IDL/LDL fractions (**E** and **F**) or chylomicron fractions (**G** and **H**) were calculated from the raw data shown in *Figure 10—figure supplements 1–H*. The differences in (**E**) did not reach statistical significance, but the changes in some subfractions in the raw data were significant. (**I** and **J**) The ratio of TG to cholesterol was calculated in chylomicron (**I**) and VLDL/IDL/LDL (**J**) fractions. (**K–M**) TG (**K**), cholesterol (**L**), and PC (**M**) concentration in apoB-containing lipoproteins precipitated from plasma of wild type and LPCAT3-deficient mice (n = 3). Values were normalized to apoB to estimate the amount of each component per particle. See also *Figure 10—figure supplement 1*.

The following figure supplement is available for figure 10:

**Figure supplement 1**. Fractionation of plasma lipoproteins by gel-filtration chromatography.

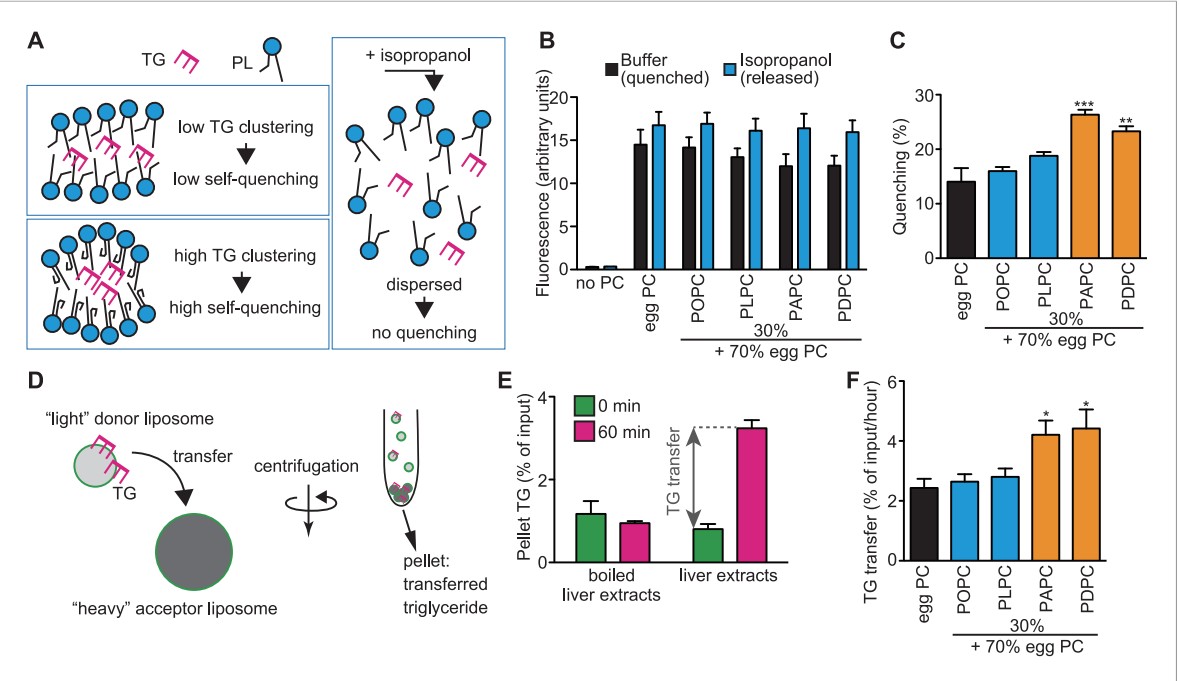

**Figure 11**. A PUFA-rich membrane promotes TG clustering and efficient transfer. (**A**) Outline of TG clustering analysis based on fluorescence quenching. When TG clustering is higher, quenching increases and fluorescence decreases. Isopropanol disrupts liposomes and TG clustering, leading to fluorescence without quenching. (**B** and **C**) Fluorescence (**B**) and the quenching rate (**C**) of NBD-TG were analyzed in PC liposomes of different compositions. (**D**) Outline of the MTP assay, measuring the transfer of NBD-TG from 'light' donor liposomes to 'heavy' acceptor liposomes. 'Heavy' acceptor liposomes are prepared in the presence of sucrose, and can be pelleted by centrifugation. (**E**) Fluorescence of NBD-TG in acceptor liposomes, after incubation for the indicated time with donor liposomes and liver extracts (or boiled extracts). NBD-TG is transferred to acceptor liposomes after incubation with liver extracts for 1 hr. The difference is used for calculation of TG transfer efficiency. (**F**) TG transfer efficiency was measured using donor PC liposomes of different compositions. Error bars are SEM (n = 3). *$p < 0.05$, **$p < 0.01$, ***$p < 0.001$. See also *Figure 11—figure supplements 1, 2*.

The following figure supplements are available for figure 11:

**Figure supplement 1**. PC acyl-chain composition of lipoproteins.

**Figure supplement 2**. Acyl-chain composition of egg PC.

Next, we analyzed whether this local pool of clustered TGs facilitates their transfer by MTP from donor liposomes to acceptor liposomes (*Figure 11D*). The acceptor liposomes, which contained sucrose and were made heavier, were separated from donors by centrifugation, and the fluorescence in the pellet was measured to detect the transfer (*Figure 11D,E*). Incubation of liver extracts with donor and acceptor liposomes for 1 hr increased the fluorescence of NBD-TG in acceptor liposomes, reflecting TG transfer by MTP activity (*Figure 11E*). Using this assay, we found that high levels of PUFAs in donor liposomes facilitate TG transfer by MTP (*Figure 11F*). Therefore, the local enrichment of membrane PUFAs by LPCAT3 might facilitate the clustering of TGs and their transport by MTP. Although LPC levels were increased in LPCAT3-deficient mice, LPC (1–5% of PC in liposome) did not affect TG clustering and transfer, at least under the present assay conditions (data not shown).

## Discussion

### Changes in the lipidome of LPCAT3-deficient mice

In this study, we first analyzed in detail how LPCAT3 affects LPLAT activity and lipid composition. Although multiple LPLATs have been identified (*Hishikawa et al., 2014*), we found that LPCAT3-deficient mice lack most of the activity required for the remodeling of PC, PE, and possibly PS (*Figures 2G, 4F*, and data not shown). The remodeling pathway has been thought to modify the

acyl-chain composition of phospholipids synthesized de novo and to be especially important for PUFA accumulation (*Hill and Lands, 1968*; *MacDonald and Sprecher, 1991*; *Lands, 2000*). Analyses of LPCAT3-deficient mice revealed complex changes in the lipidome, but among the major fatty acids, the most commonly decreased one was arachidonate (*Figures 3–5*). Our results show that the remodeling by LPCAT3 is highly selective for arachidonate accumulation and that many PUFA species persist in membranes of LPCAT3-deficient mice (*Figures 3–5*). Therefore, the remodeling by LPCAT3 does indeed modulate the acyl-chain composition of de novo synthesized phospholipids, but is not completely requisite for PUFA accumulation. The observation that C22 and C24 n-6 PUFAs that arose after arachidonate elongation accumulate in LPCAT3-deficient mice (*Figure 3—figure supplement 2*) shows that this enzyme utilizes arachidonoyl-CoA competitively with other enzymes. C22 n-6 PUFAs increased not only in phospholipids but also in neutral lipids (*Figure 7B*). Since TG and phospholipids share phosphatidic acid as the same precursor (*Coleman and Lee, 2004*), this suggests that in LPCAT3-deficient mice, the C22 n-6 PUFA-CoAs are utilized by LPAATs during de novo synthesis, together with one part of the excess of arachidonoyl-CoA (*Figure 12*). The unchanged docosahexaenoate in neutral lipid fractions (*Figure 7B*) suggests that its increase in phospholipids (*Figure 7A*) is not due to an accumulation by LPAATs, but rather suggests that docosahexaenoate-containing phospholipids were

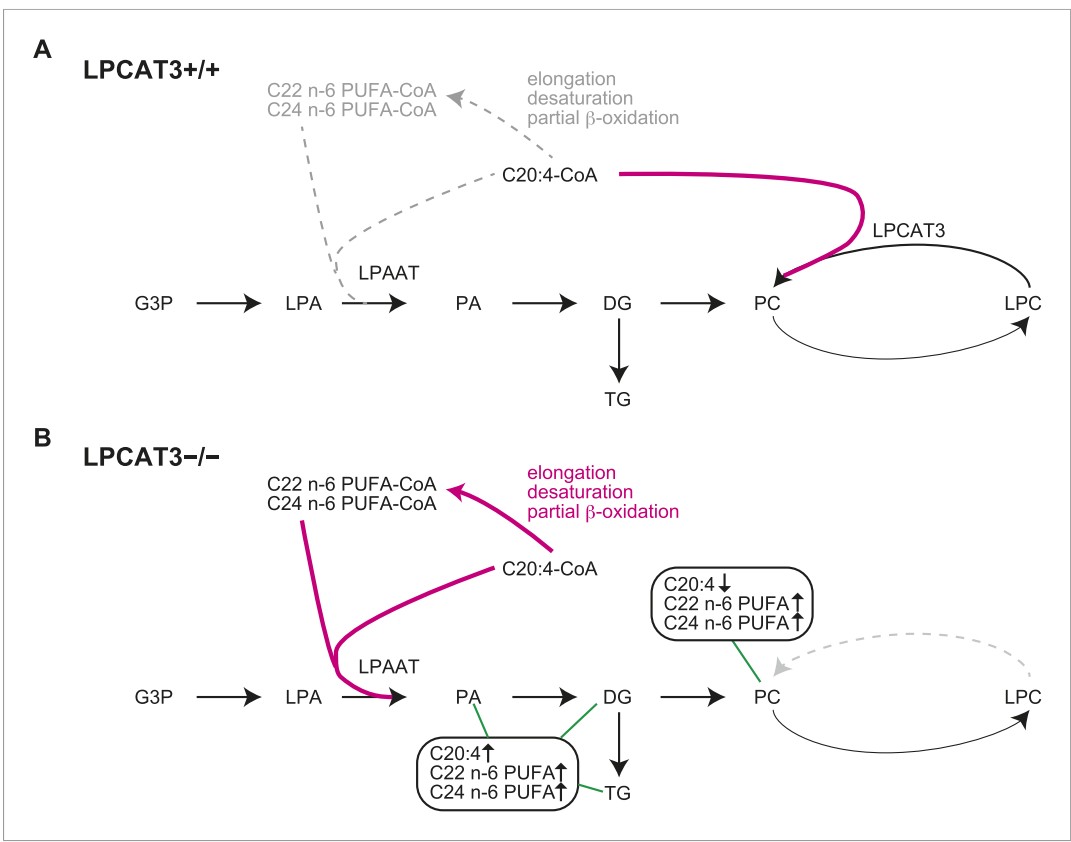

**Figure 12**. Proposed model of arachidonoyl-CoA shunting in LPCAT3-deficient mice. During de novo synthesis, TG and PC share the same precursor, diacylglycerol (DG). Therefore, the fatty acid profile of neutral lipids is suggestive of incorporation during de novo synthesis of both TG and phospholipids, assuming that TG has a negligible degree of acyl-chain remodeling. Based on the enzymatic assays and the fatty acid profiles of phospholipids and neutral lipids, we propose the following explanation for the results. (**A**) In wild type mice, arachidonoyl-CoA is largely utilized by LPCAT3 to be incorporated into phospholipids. (**B**) In LPCAT3-deficient mice, an excess of arachidonoyl-CoA occurs, which is utilized by LPAATs for de novo synthesis of both phospholipids and triglycerides, either directly or after being metabolized into C22 and C24 n-6 PUFAs (*Figure 3—figure supplement 2*). This pathway is less utilized in wild type mice. The monoacylglycerol acyltransferase pathway of TG synthesis is ignored in this figure, since differences in TG profiles should originate from those synthesized by the embryo and not those provided by the mother bloodstream. G3P: glycerol 3-phosphate; LPA: lysophosphatidic acid; PA: phosphatidic acid.

not utilized for acyl-chain remodeling in LPCAT3-deficient mice. Therefore, we propose that the increased n-6 C22 PUFAs and C24 PUFAs in phospholipids of LPCAT3-deficient mice are due to increased incorporation during de novo synthesis (*Figure 12*), although we cannot exclude the possibility that unknown LPCAT enzymes selective for these PUFAs exist. In addition to the changes in phospholipid acyl-chain composition, LPCAT3 deficiency induced changes in LPC (in liver and small intestine) or total PC (in proximal small intestine) levels, but only in tissues where LPCAT3 expression is high (*Figures 2B, 3A,H*). LPCAT3 deficiency might have caused a pronounced imbalance between LPC and PC in these tissues.

It is of note that when comparing different tissues, the extent of arachidonate decrease in LPCAT3-deficient mice does not correlate with the expression levels of this enzyme (*Figures 2A, 5A*). Even in tissues such as the brain and heart, where LPCAT3 expression is relatively small, arachidonate levels were prominently reduced in LPCAT3-deficient mice. Therefore, LPCAT3 is required for accumulating arachidonate, but the expression level of this enzyme is not the single factor to explain different arachidonate levels in tissues. Rather, the different levels should be explained by both LPCAT3 activity and supply of arachidonoyl-CoA, suggesting that tissue arachidonate levels are regulated in the remodeling pathway with dependence on substrate supply, as we recently proposed (*Harayama et al., 2014*). In addition, the observation that eicosanoids are unchanged in LPCAT3-deficient mice (*Figure 6*) suggests that the phenotypes of LPCAT3-deficient mice are unrelated to these lipid mediators. Therefore, analyses of LPCAT3-deficient mice revealed novel functions of membrane arachidonate that are not attributed to eicosanoids. In the future, it is of interest to determine eicosanoid levels in stimulated cells, where phospholipase $A_2$ is activated (*Shimizu, 2009*; *Dennis et al., 2011*). In addition, since changes in linoleate and docosahexaenoate levels were also observed in some tissues (*Figure 5B,C*), it will be interesting to investigate other lipid mediators, such as hydroxyoctadecadienoic acids, resolvins, and protectins (*Obinata et al., 2005*; *Buckley et al., 2014*).

## Mechanism of lipid droplet accumulation in LPCAT3-deficient mice

We next showed that LPCAT3-deficient mice accumulate cytosolic lipid droplets (*Figures 7, 9*) in hepatocytes and enterocytes due to an inefficiency of luminal TG transport by MTP. The quenching assay (*Figure 11C*) suggests that this inefficiency is caused by the differences in TG clustering, which is facilitated by the enrichment of PUFAs in a local membrane by LPCAT3 (*Figure 13A*). In this section, we will describe how we reached this conclusion from the literature and our results.

Intestine-specific MTP deficiency leads to (at least partial) neonatal lethality, which was explained by lipotoxicity (see the tamoxifen-treated apoB48 *Mttp*-IKO mice in [*Xie et al., 2007*]). LPCAT3-deficient mice display similar lethality and enterocyte lipid accumulation, and have less TGs in plasma lipoproteins (*Figures 8–10*). Therefore, we attribute the lethality of LPCAT3-deficient mice to the intestinal lipotoxicity caused by the intake of large amounts of milk TG under MTP inefficiency. Indeed, enterocyte damage was obvious in LPCAT3-deficient mice, and probably resulted in an inability to absorb nutrients, not limited to sugars and lipids. One difference between LPCAT3-deficient mice and MTP-deficient mice is that the former assemble cholesterol normally in lipoproteins (*Figure 10F,H*). This suggests that LPCAT3 deficiency does not directly inhibit MTP, but may affect the properties of the MTP substrate TG. The quenching assay supports this, and one of these properties is the clustering of TG between leaflets (*Figure 11C*). It is of note that TG absorption is normal in LPCAT3-deficient mice at P0.4 (*Figure 10G*). This suggests that the difference in transport efficiency is apparent only when TGs are in high amounts and require to be densely clustered in the membrane for efficient transport. Therefore, it is possible that cholesterol absorption is also affected when feeding a high cholesterol diet.

Comparison with reports of LPCAT3 knockdown in liver led us to focus on the synthesis of PUFA-containing phospholipids in a 'local membrane microenvironment'. In a previous report, LPCAT3 knockdown did not impair TG transport, but rather increased VLDL secretion by the liver, due to the induction of MTP caused by increased LPC (*Li et al., 2012*). Despite this opposite result, the changes in the lipidome had similarities; LPC increased and arachidonate-containing PC decreased (*Figure 3*) (*Li et al., 2012*; *Rong et al., 2013*). Therefore, these similar changes cannot explain the impaired TG transport in LPCAT3-deficient mice. LPCAT3 is ideal to generate a local high PUFA concentration, since the enzyme is very selective for linoleoyl- and arachidonoyl-CoA (*Figures 1B, 13A,B*) (*Hishikawa et al., 2008*; *Zhao et al., 2008*). On the other hand, the enzyme catalyzing the last step of PC de novo synthesis, diacylglycerol cholinephosphotransferase (CPT) is not selective for diacylglycerol species

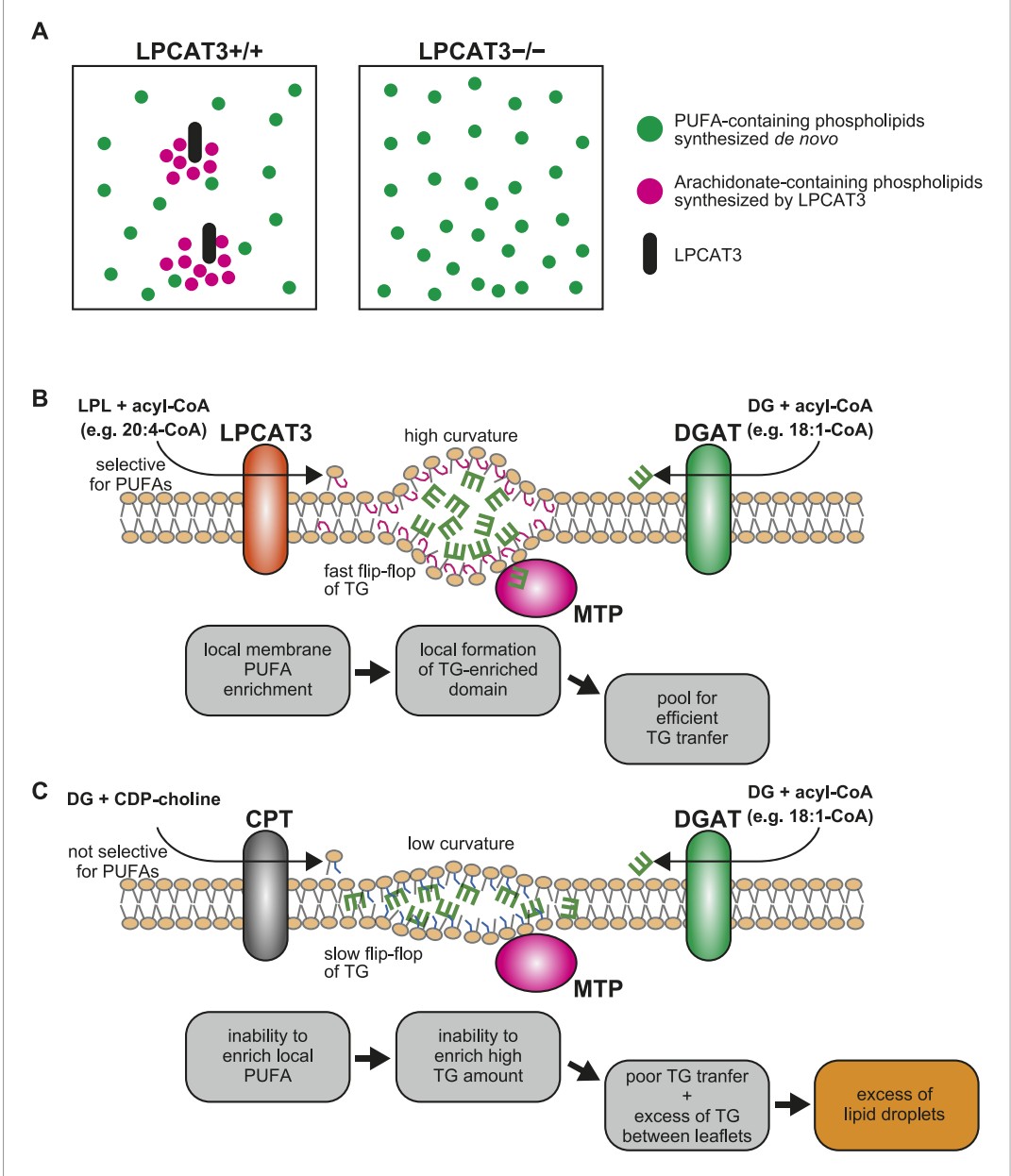

**Figure 13**. A proposed mechanism of TG transport facilitated by LPCAT3. (**A**) Differences in PUFA-containing phospholipids distribution between wild type and LPCAT3-deficient mice. Since PUFAs are shunted into de novo synthesis (*Figure 12*), the total levels of PUFAs are similar in both genotypes. However, since the remodeling by LPCAT3 is highly selective for arachidonate, arachidonate-containing phospholipids are concentrated around LPCAT3, generating a local membrane area rich in PUFAs. (**B** and **C**) Proposed mechanism of TG transport facilitated by LPCAT3. (**B**) LPCAT3 locally enriches PUFA-containing phospholipids. When TG synthesis occurs in the proximity, this PUFA-enriched domain favors the formation of a blister-like structure with a high capacity and high surface curvature. TG in the blister-like structure has a high flip-flop rate, enabling efficient supply to MTP at the luminal side. (**C**) The substrate selectivity of the final step of PC synthesis does not favor the generation of a PUFA-enriched domain. LPL: lysophospholipid; DG: diacylglycerol; DGAT: diacylglycerol acyltransferase; CPT: diacylglycerol cholinephosphotransferase.

containing PUFA (*Mantel et al., 1993*), and thus is unable to do the same (*Figure 13C*). Therefore, local PUFA-rich domains arise continuously around LPCAT3 in wild type mice, but not in LPCAT3-deficient mice (*Figure 13A*). Residual LPCAT3 in knockdown experiments might be sufficient for this

local PUFA enrichment. The results of *Figure 11* suggest that if TG synthesis by diacylglycerol acyltransferases occurs in proximity of a local PUFA-rich domain, TG clustering will occur, providing a pool for efficient transport by MTP (*Figure 13B*). Although further studies are required for a complete understanding, the different results between the LPCAT3 knockdown study (*Li et al., 2012*; *Rong et al., 2013*) and our present study might be due to the ability to accumulate PUFAs locally. The observation that not only arachidonate but also docosahexaenoate promote TG clustering and transfer, as well as the fact that total PUFA levels are not largely changed in LPCAT3-deficient mice, again suggest the necessity of considering a local high concentration. In the presence of LPCAT3, PUFAs might be unevenly distributed in the membrane, and this should be required to reach a sufficient high PUFA concentration for efficient TG clustering and transfer. Therefore, although docosahexaenoate can substitute arachidonate in in vitro TG transfer assays, its inability to be locally concentrated by LPCAT3 make this fatty acid incompetent for lipoprotein assembly.

Other mouse models that have abnormal PUFA metabolism are mice that are deficient in fatty acid desaturase 1 (FADS1) or FADS2. These enzymes induce double bonds at different steps of PUFA generation, and mice lacking either gene encoding them could not synthesize arachidonate from linoleate. Knockout mice of both FADS1 and FADS2 have been generated, and displayed multiple abnormalities such as lethality, decreased intestinal crypt proliferation, and sterility (*Stoffel et al., 2008*; *Stroud et al., 2009*; *Fan et al., 2012*). However, the accumulation of intestinal lipid droplets was not reported. It is possible that these mice might receive enough arachidonate from breast milk during the lactating period, making it difficult to analyze intestinal arachidonate functions at the perinatal period. Indeed, most studies report abnormalities of FADS1/2-deficient mice after weaning. We speculate that LPCAT3 is required for intestinal TG absorption especially when TG levels in food are high, as is the case in milk. Therefore, it is intriguing to observe phenotypes by feeding these mice with high-fat diet. It is of note that lipid droplet accumulation in the liver is seen in FADS2-deficient mice, and that arachidonate feeding reverses this phenotype (*Stoffel et al., 2014*), which may fit with our proposed model. It will be important to compare these different models in detail to unveil more functions of membrane PUFAs in future studies.

## Mechanism of TG clustering by local PUFA enrichment

The quenching assay shows that a PUFA-rich membrane enables a dense clustering of TGs. This composition may favor the deformation of the membrane surrounding TGs in a 'blister-like' morphology, with high local curvature (*Khandelia et al., 2010*). When compared with a membrane rich in monounsaturated fatty acids, a PUFA-rich membrane is more easily deformed when it is bent (*Vanni et al., 2013*; *Pinot et al., 2014*). Based on this theory, it can be imagined that when TG synthesis occurs intensively, there is a demand to generate a highly curved blister-like structure to store more TGs in a limited place for efficient transport by MTP, and that a PUFA-rich environment generated by LPCAT3 helps this process by making the membrane more flexible and mobile (*Figure 11A*).

Our model suggests that one benefit of regulating acyl-chain composition is to adapt membrane flexibility to a required curvature. When we analyzed plasma lipoprotein PC, we detected many monounsaturated species (*Figure 11—figure supplement 1*). This composition is similar to that reported for PC surrounding cytosolic lipid droplets (*Tauchi-Sato et al., 2002*). Therefore, in the case of a membrane surrounding TGs with a large diameter, the composition is shifted to one with more monounsaturated species, which makes the membrane less flexible. This suggests that membranes surrounding TGs acquire acyl-chains that fit well with the curvature required.

It is unknown how the local clustering affects TG transfer. Molecular dynamics simulations suggested that when TG is clustered, the efficiency of its flip-flop between leaflets is accelerated (*Khandelia et al., 2010*). Since MTP is present only in the luminal side of the bilayer, it is likely that the accelerated flip-flop enables TG to be present in the leaflet that is accessible by MTP (*Figure 13B*). Future studies would clarify how MTP is acting at the membrane interface to answer how membrane PUFAs enable efficient TG transfer.

## Conclusions and perspective

In cases where many lipids are ingested (as in the neonatal period), the efficient export of TG is important not only for nutrition, but also to prevent lipotoxicity on enterocytes (*Xie et al., 2007*). The generation of an arachidonate-enriched membrane microenvironment by LPCAT3 might be required for efficient clustering of TGs and their correct transport (*Figure 13B*). Previously, the functions of

membrane arachidonate that are not related to eicosanoids were largely unknown. In addition, the mechanism and factors that enable MTP to efficiently transfer TGs unidirectionally into the ER lumen were poorly understood. Our study reveals for the first time that membrane arachidonate and TG clustering are critical factors to regulate the correct luminal directionality of TG transport, which has lethal consequences when improperly regulated. Therefore, the present study identifies LPCAT3 as a novel key molecule involved in normal lipoprotein assembly. During the final editing of the revised manuscript, Rong et al. reported LPCAT3-deficient mice with similar conclusions. In the future, conditional ablation of LPCAT3 in different contexts will reveal other functions of arachidonate in phospholipids and will provide further critical insights into membrane biology.

## Materials and methods

### Expression plasmids

Expression plasmids for FLAG-LPCAT3 and FLAG-LPCAT3 H374A were generated previously in a pCXN2 expression vector, which expresses the inserted construct under the control of a CAG promoter and contains a neomycin-resistance gene (*Niwa et al., 1991*; *Hishikawa et al., 2008*; *Shindou et al., 2009*).

### Construction of single guide RNA expression vectors

The empty vector for the expression of single guide RNAs, pX459 was obtained from Addgene. Oligos that are listed in a supplementary table were annealed and inserted into the Bbs I sites of pX459 (Addgene, Cambridge, MA, plasmid 48139) by Golden Gate assembly (*Engler et al., 2008*) as previously described (*Ran et al., 2013*), using Bpi I (Thermo Fisher Scientific K.K., Kanagawa, Japan) and T4 DNA ligase (New England Biolabs, Ipswich, MA). Plasmids were transformed in ECOS competent *Escherichia coli* JM109 (Nippon Gene Co., Ltd., Toyama, Japan). Sequences were verified by a sequencing service (Eurofins Genomics K.K., Tokyo, Japan), and plasmids were purified using a Plasmid Midi Kit (Qiagen K.K., Tokyo, Japan).

### Hepatoma cell culture, transfection, and genome editing

RH 7777 cells were cultured on IWAKI collagen-coated dishes (Asahi Glass Co., Ltd., Tokyo, Japan) in Dulbecco's Modified Eagle Medium (Nacalai Tesque, Inc., Kyoto, Japan) supplemented with 10% GIBCO fetal bovine serum (Life Technologies Japan Ltd., Tokyo, Japan), in a humidified incubator at 37°C with 5% $CO_2$. For lipid analysis, cells were cultured for 24 hr in the presence of 10 μM each of linoleate, arachidonate, and docosahexaenoate (Cayman Chemical Company, Ann Arbor, MI) to make the changes in PUFA-containing PC more easily distinguished (although most of the changes reported in the manuscript can be seen without this supplementation). Lipofectamine 3000 (Life Technologies) was used for transfection. For the establishment of stable transfectants, cells were selected in 2 mg/ml G418 (Life Technologies) for 1 week, starting at 24 hr post transfection. Selected cells were maintained in medium containing 0.3 mg/ml G418. LPCAT3-null cells were established by transfecting a pair of single guide RNAs, inserted in the expression vector pX459 (Addgene, plasmid 48,139). Transfection with this vector leads to the simultaneous expression of single guide RNAs, Cas9, and a puromycin resistance gene. Single guide RNAs were designed to flank the region coding the WHG sequence of rat LPCAT3 and cause a ~100 bp deletion in *Lpcat3* gene (*Figure 1—figure supplement 3A*). The WHG sequence is conserved in all members of the membrane-bound *O*-acyltransferase family (*Hofmann, 2000*), and is required for LPCAT3 activity (*Shindou et al., 2009*). Cells that were transfected transiently were selected by puromycin (10 μg/ml, InvivoGen, San Diego, CA) for 24 hr. Clones were obtained by limiting dilution on 96 well plates. Clones that contained deletion in the *Lpcat3* locus were screened by PCR using genomic DNA of each clone as a template, and ExTaq HS DNA polymerase (Takara Bio Inc., Shiga, Japan) (*Figure 1—figure supplement 3A*).

### Preparation of protein samples

Cultured cells were scraped in ice-cold T20 buffer (20 mM Tris-HCl [pH7.4, Wako Pure Chemical Industries, Ltd., Osaka, Japan], 300 mM sucrose [Wako], and a proteinase inhibitor mixture, Complete [Roche Diagnostics K.K., Tokyo, Japan]) and sonicated using a probe sonicator (Ohtake Works, Tokyo, Japan). Frozen tissues (1–100 mg) were homogenized in ice-cold T100 buffer (100 mM Tris-HCl (pH7.4), 300 mM sucrose, and Complete) using a Physcotron homogenizer (Microtec Co. Ltd., Chiba,

Japan). The homogenate was centrifuged at 800×g for 10 min. The supernatant was used for western blot analysis of MTP and PDI. The same supernatant was centrifuged at 100,000×g for 1 hr to obtain membrane fractions. The pellet was resuspended in TSE buffer (20 mM Tris-HCl [pH7.4], 300 mM sucrose, and 1 mM EDTA [Dojindo Laboratories, Kumamoto, Japan]). This membrane fraction was used for western blot analysis of LPCAT3 and the enzymatic assays. Protein concentration was measured using the Bio-Rad Protein Assay (Bio-Rad Laboratories, Inc., Hercules, CA). Protein samples were snap frozen in liquid nitrogen and stored at −80°C until use.

## Western blot analysis

Protein samples were resolved on 10% SDS-polyacrylamide gels and electrophoretically transferred to nitrocellulose membranes (GE Healthcare UK Ltd., Buckinghamshire, England) using a Trans-Blot SD semi-dry transfer cell (Bio-Rad). Membranes were stained with Ponceau S (Sigma–Aldrich Co. LLC., St. Louis, MO), and then blocked overnight with 5% skim milk (BD Biosciences, Franklin Lakes, NJ) in Tris-buffered saline with 0.1% Tween 20 (Wako) (TBST). Primary antibodies were diluted in 5% skim milk/TBST as following: anti-LPCAT3 (40 ng/ml), anti-FLAG M2 antibody (5 µg/ml, Sigma–Aldrich), anti-MTP antibody (50 ng/ml, BD), or anti-PDI antibody (1:1000 dilution, Cell Signaling Technology, Inc., Danvers, MA). Horseradish peroxidase-conjugated secondary antibodies (GE Healthcare) were used at a 1:2000 dilution in 5% skim milk/TBST. TBST was used for washing steps and changed at least three times between incubation steps. ECL select western blot detection system (GE Healthcare) was used for chemiluminescence, and detected using ImageQuant LAS500 (GE Healthcare).

## Enzymatic assays

LPCAT assays were performed as previously described, in a condition that provides linearity (*Harayama et al., 2014*; *Martin et al., 2014*). Briefly, membrane proteins (0.01 µg/tube) were mixed with 25 µM deuterium-labeled 16:0 LPC or non-labeled 17:1 LPE and 1 µM each of 16:0-, 18:1-, 18:2-, 20:4-, and 22:6-CoA at 37°C for 10 min (all from Avanti Polar Lipids, Inc., Alabaster, AL). Reaction mixtures contained 110 mM Tris-HCl (pH 7.4), 1.5 mM EDTA, 2 mM $CaCl_2$ (Wako), 0.015% Tween-20, and 150 mM sucrose in a total volume of 100 µl. Reactions were stopped by the addition of 300 µl chloroform/methanol (1/2, Wako). Internal standards (dilauryl-PC for LPCAT assay and dimyristoyl-PE for LPEAT assay, Avanti) were added, lipids were extracted by the method of Bligh and Dyer (*Bligh and Dyer, 1959*), dried using a centrifugal evaporator (Sakuma Seisakusho Ltd., Tokyo, Japan), and reconstituted in methanol. Products were measured by LC-MS. Products were separated on an ACQUITY UPLC BEH C8 column (1.7 µm, 2.1 × 30 mm, Waters Corporation, Milford, MA) using a linear gradient of solvent B (acetonitrile, Wako) over solvent A (20 mM $NH_4HCO_3$/water, Wako), using an ACQUITY ultra performance liquid chromatography (UPLC) system (Waters). The flow rate was 800 µl/min. The gradient started at 55% solvent B, was linearly increased to 95% solvent B in 4.5 min, and maintained for 1.5 min. Detection was done on a TSQ Vantage triple stage quadrupole mass spectrometer (Thermo Fisher Scientific) by selected reaction monitoring (SRM). Transitions were $[M + H]^+ \rightarrow 184.1$ for PC and $[M + H]^+ \rightarrow [M + H-141]^+$ for PE (both in the positive ion mode electrospray ionization). Signals of LPCAT products were compared to calibration curves of nonlabeled standards for quantification. When total LPCAT activity is shown, it illustrates the sum of the five products generated during the assay. Standards for LPEAT products were not available, and peak areas normalized to the internal standard were used as a measure of enzymatic activity.

## Lipid analysis of cultured cells, precipitated lipoproteins, and egg PC

Lipids from cultured cells were obtained using methanol. Lipids from precipitated lipoproteins were extracted by the method of Bligh and Dyer. Egg PC (Sigma–Aldrich) in ethanol (Wako) was used after dilution in methanol. Lipids were separated on an ACQUITY UPLC BEH C8 column (1.7 µm, 1.0 × 100 mm) using a linear gradient of solvent B (acetonitrile) over solvent A (20 mM ammonium bicarbonate), using an ACQUITY UPLC system. The flow rate was 100 µl/min. The gradient started at 20% solvent B, was linearly increased to 95% solvent B in 20 min, and maintained for 15 min. PC was detected using a precursor ion scanning for *m/z* 184.1 in the positive ion mode electrospray ionization using a TSQ Vantage triple stage quadrupole mass spectrometer. This method does not resolve the acyl chains at the *sn*-1 and *sn*-2 positions. To characterize the acyl-chain composition of selected peaks, SRM was additionally performed in the negative ion mode. The transitions were $[M + HCO_3]^- \rightarrow 255.2$ for species containing palmitate, and $[M + HCO_3]^- \rightarrow 283.2$ for those containing stearate. Although this

method is more specific in the discrimination of acyl-chains, we obtained a better linear dynamic range in the positive ion mode, and thus used the precursor ion scanning for the figure in this manuscript. Peak areas of all detected diacyl-PC species were summed, and the ratio of individual species (as % of total) was calculated. See *Figure 1—figure supplement 2* for an example of peak annotation using different detection methods.

## Annotation of phospholipids

When using LC-MS methods that detect only the sum of both *sn*-1 and *sn*-2 fatty acids in phospholipids, the molecule is illustrated as XX:Y, where XX is carbon number and Y is double bond number, as a sum of both acyl-chains. When additional detection of fatty acid fragments is performed to confirm the acyl-chain composition, the molecule is illustrated as AA:B-CC:D, where AA and CC are carbon numbers, and B and D are double bond numbers. Note that the methods do not discriminate which *sn*- position each fatty acid resides (meaning that the above phospholipid might also have been CC:D-AA:B).

## Animals

All animal experiments were approved by and performed in accordance with the guidelines of the Animal Research Committee of National Center for Global Health and Medicine (12,053, 13,009, 14,045), and the animal experimentation committee of the University of Tokyo (H09-144, P08-042).

## Tissue sampling from mice

Newborn mice and embryos were euthanized by rapid decapitation and had laparotomy. Tissues were harvested and were snap frozen in liquid nitrogen and stored at −80°C until RNA and protein extraction. For histological analyses, tissues were fixed as described in each section.

## Isolation of total RNA and real-time quantitative PCR

Frozen tissues (0.5–100 mg) were homogenized using a Physcotron homogenizer in QIAzol Lysis Reagent (Qiagen) and total RNA was extracted using an RNeasy Mini Kit (Qiagen). Complementary DNA synthesis was carried out using SuperScript III reverse transcriptase (Life Technologies) using 1 μg total RNA as a template. Real-time quantitative PCR was performed using Fast SYBR Green Master Mix and the Step One Plus real-time PCR system (Life Technologies), using the primers listed in a supplementary table.

## LPCAT3 antibody production

Anti-LPCAT3 antibody was produced by Sigma–Aldrich. Rabbits were immunized with a C-terminal LPCAT3 peptide (CHKAMVPRKEKLKKRE). Blood was collected and antiserum was obtained. The same C-terminal peptide was immobilized on activated thiol-Sepharose 4B (GE Healthcare), and anti-LPCAT3 antibody was affinity purified using this column.

## Generation of LPCAT3-deficient mice

LPCAT3-floxed mice (Accession No. CDB0653K: http://www.cdb.riken.jp/arg/mutant%20mice%20list.html) were generated as described (http://www.cdb.riken.jp/arg/Methods.html) using the HK3i embryonic stem cell line (*Kiyonari et al., 2010*). To generate a targeting vector, genomic fragments of the *Lpcat3* locus were obtained from the RP23-388D4 BAC clone (BACPAC Resources Children's Hospital Oakland, St. Oakland, CA). A 903 bp region containing exons 10, 11, and 12 of the *Lpcat3* gene was flanked by loxP sites (*Figure 2—figure supplement 1A*). Targeted ES clones were microinjected into ICR 8-cell stage embryos, and injected embryos were transferred into pseudopregnant ICR females. The resulting chimeras were bred with C57BL/6 mice, and heterozygous offsprings were identified by PCR. Exons 10 to 12 of *Lpcat3* were removed by mating these mice with C57BL/6 telencephalin-Cre mice (*Nakamura et al., 2001*; *Fuse et al., 2004*). For genotyping, DNA was extracted from tail tips and subjected to PCR using Ex Taq HS DNA polymerase (Takara). Primer sequences are described in a supplementary table.

## Measurement of phospholipids from tissues

Frozen tissues (0.5–100 mg) were pulverized and 0.8 ml methanol was added. The tissue suspensions were centrifuged at 15,000×*g*, and the collected supernatant was used for the

measurement of phospholipids. PC concentration was measured by an enzyme-based fluorescent assay with minor modifications from a previous report (*Morita et al., 2010*). Samples were dried in a centrifugal evaporator and dissolved in 0.16 ml 1% Triton X-100 (Wako) per mg wet tissue weight. 10 µl of sample was reacted with 40 µl mixture C1 (PC-Specific Phospholipase D [1:200 dilution, Cayman], 1.5 mM $CaCl_2$, 50 mM NaCl [Wako], and 50 mM Tris-HCl [pH 7.4]) at 37°C for 30 min. After generation of free choline from PC, 50 µl mixture C2 (4 U/ml choline oxidase from *Alcaligenes sp.* [Wako], 5 U/ml horseradish peroxidase [Oriental Yeast Co., ltd., Tokyo, Japan], 0.3 mM Amplex Red (Life Technologies), 0.2% Triton X-100, 50 mM NaCl and 50 mM Tris-HCl [pH7.4]) were added and incubated at room temperature for 30 min. The hydrogen peroxide generated by choline oxidase was fluorescently detected using Amplex Red reagent and measured with ARVO X3 (PerkinElmer Inc., Waltham, MA). For phospholipid analysis, the methanol extracts were further diluted with methanol to adjust the concentration to 10 mg tissue/ml (liver, brain, and lung), 7 mg tissue/ml (kidney), 5 mg tissue/ml (proximal small intestine), 3.5 mg tissue/ml (heart), 3 mg tissue/ml (distal small intestine and stomach), 1.5 mg tissue/ml (thymus), 1 mg tissue/ml (colon), and 0.5–2.7 mg tissue/ml (spleen). Embryonic spleens were very small with high variability, and were used without adjustment of concentration. This might have led to the difference in total signal in *Figure 3—figure supplement 1*. LC-SRM-MS analysis was performed using a Nexera UHPLC system and triple quadrupole mass spectrometers LCMS-8040 or LCMS-8050 (Shimadzu Corporation, Kyoto, Japan). For phospholipid analysis, an Acquity UPLC BEH C8 column (1.7 µm, 2.1 mm × 100 mm, Waters) was used with the following ternary mobile phase compositions: 5 mM $NH_4HCO_3$/water (mobile phase A), acetonitrile (mobile phase B), and isopropanol (mobile phase C). Pump gradient [time (%A/%B/%C)] was programmed as follows: 0 min (75/20/5)–20 min (20/75/5)–40 min (20/5/75)–45 min (5/5/90)–50 min (5/5/90)–55 min (75/20/5). Flow rate was 0.35 ml/min and column temperature was 47°C. Injection volume was 5 µl. SRM analysis with phospholipid class discrimination was performed with the following transitions: $[M + H]^+ \rightarrow 184$ for PC, $[M + H]^+ \rightarrow [M + H-141]^+$ for PE, $[M-H]^- \rightarrow [M-H-87]^-$ for PS, $[M-H]^- \rightarrow 241$ for PI. Peak areas of all the detected species were summed to obtain the total signal. Peak areas of individual species were normalized with this sum, and are illustrated as % of total. For the comparison of *Figure 3B*, PC species that were detected in all samples were selected. The ratio between LPCAT3-deficient mice and wild type mice (as % of wild type) was calculated for each species in every tissues. These values were further averaged for all tissues to detect the changes that occurred globally. As in the case of lipid analysis for cultured cells, this method does not resolve the acyl chains at the *sn*-1 and *sn*-2 positions. For relevant phospholipid species, additional SRM analyses with selection of fatty acid fragments at Q3 were performed to confirm that they contain the fatty acid of interest. Detection was performed at the negative ion mode using the following transitions: $[M + HCO_3]^- \rightarrow [FA-H]^-$ for PC, $[M-H]^- \rightarrow [FA-H]^-$ for PE, where [FA] is the monoisotopic mass of the fatty acid of interest. We set up SRM channels for all possible acyl-chain composition, based on the assumption that the following fatty acids are present in vivo: C12:0, C12:1, C14:0, C14:1, C16:0, C16:1, C17:0, C17:1, C18:0 to C18:3, C20:0 to C20:5, C22:0 to C22:6, C24:0 to C24:6. The data presented in the manuscript are derived from SRM channels detecting the fatty acid fragment with more unsaturation (or the longer one when both have the same unsaturation), but only molecules that were detected in SRM channels for both fatty acid fragment with consistent retention times were analyzed.

## Measurement of fatty acids by GC-FID

For fatty acid analysis by GC-FID, lipid samples were extracted from tissues by the method of Bligh and Dyer. C23:0 (Supelco n-Tricosanoic acid, Sigma–Aldrich) was added to the extracted samples as an internal standard. Then, the fatty acids were methylated with the Fatty Acid Methylation Kit (Nacalai Tesque), and purified using the Fatty Acid Methyl Ester Purification Kit (Nacalai Tesque) following the manufacturer's instructions. Fatty acid methyl ester samples were characterized with GC-2010 Plus system (Shimadzu) equipped with an FID. The flow rate of carrier gas (He) was set at 45 cm/s linear velocity. The temperature of the injection unit and the detector were 240°C and 250°C, respectively. The oven temperature was initiated at 140°C, then raised to 200°C at a rate of 11°C/min, then increased to 225°C at a rate of 3°C/min, and finally elevated to 240°C at a rate of 20°C/min and held at this temperature for 5 min. The injection volume was 2 µl in the split injection mode. For separation, a capillary column (FAMEWAX, 30 m, 0.25 mm ID, 0.25 µm; Restek Corporation,

Bellefonte, PA) was used. Fatty acid methyl esters were identified and quantified using a mixture of fatty acid methyl ester standards (Supelco 37 Component FAME Mix and DPA (n-3) from Sigma–Aldrich; DPA (n-6) from Nu-Chek Prep, Inc., Elysian, MN; DTA (n-6) from Cayman) for calibration.

## Lipid fractionation

Neutral lipids and phospholipids were separated by solid phase extraction using InertSep NH2 aminopropyl columns (GL Sciences Inc., Tokyo, Japan). Lipids were extracted using the method of Bligh and Dyer, dried in a centrifugal evaporator, dissolved in chloroform, and applied to the columns. Flow-through was collected and combined with the fraction eluted by chloroform:isopropanol (2:1 by volume) as the neutral lipid fraction. After the removal of free fatty acids by 2% acetic acid in diethyl ether, phospholipids were eluted using 2.8% ammonia in methanol.

## Measurement of eicosanoids

For quantification of eicosanoids, internal standards were spiked in methanol extracts, and samples were purified with solid phase extraction with an Oasis HLB column (Waters) as previously described (*Kita et al., 2005*). Eicosanoids were quantified by a triple quadrupole mass spectrometer LCMS-8040 (Shimadzu). Separation was performed on a Kinetex C8 column (2.6 µm, 2.1 × 150 mm, Phenomenex) with a binary mobile phase of the following compositions: 0.1% formic acid/water (mobile phase A) and acetonitrile (mobile phase B). Pump gradient [time (%A/%B)] was programmed as follows: 0 min (90/10)–5 min (75/25)–10 min (65/35)–20 min (25/75)–20.1 min (5/95)–28 min (5/95)–28.1 min (90/10)–30 min (90/10). Flow rate was 0.4 ml/min and column temperature was 40°C. SRM transitions were: $369.3 \rightarrow 245.2$ for 6-keto-PGF$_1\alpha$, $373.3 \rightarrow 249.2$ for 6-keto-PGF$_1\alpha$-d4, $351.2 \rightarrow 271.2$ for PGE$_2$, $355.2 \rightarrow 275.2$ for PGE$_2$-d4, $319.2 \rightarrow 219.2$ for 15($S$)-HETE, and $327.2 \rightarrow 226.2$ for 15($S$)-HETE-d8. Signals were compared to those of standard curves for quantification as previously described (*Kita et al., 2005*).

## Measurement of TG and cholesterol in plasma and tissue

Tissue lipids were obtained as described above in the lipid fractionation (before column separation). The dried lipid samples were dissolved in 5% Nonidet P-40 (Nacalai tesque), and heated at 95°C for 5 min followed by vortexing. This process was repeated to increase the solubility of neutral lipids. TG and cholesterol levels were measured using LabAssay Triglyceride Kit and LabAssay Cholesterol Kit (Wako), respectively.

## Histological analysis of paraffin or frozen sections

Collected tissues were fixed using 10% neutral buffered formalin (Wako) or 4% paraformaldehyde (PFA, Wako) for 24 to 48 hr. After fixation, tissues were embedded in Tissue-Tek paraffin wax II 60 (Sakura Finetek Japan, Tokyo, Japan) and sectioned by a sliding microtome REM-700 (Yamato Kohki Industrial, Saitama Japan). Otherwise, tissues were embedded into NEG50 frozen section medium (Thermo Scientific), frozen, and sectioned by Cryostat CM3050 S (Leica Biosystems GmbH, Nussloch, Germany). The following dyes were used for staining the sections: hematoxylin (Wako)/eosin (Muto Pure Chemicals Co., LTD., Tokyo, Japan), Periodic acid-Schiff (Muto Pure Chemicals)/hematoxylin (Wako), or oil red O (Sigma–Aldrich)/hematoxylin (Vector). Some of the microscopic examinations were performed at BoZo Research Center Inc. Tokyo, Japan.

## Electron microscopy

Samples were pre-fixed with 2% PFA and 2% glutaraldehyde (GA, in 30 mM HEPES buffer (pH7.4)) for overnight at 4°C, followed by post-fixation with aldehyde–OsO$_4$ mixture (1.25% GA, 1% PFA, 0.32% K3[Fe(CN)$_6$], and 1% OsO$_4$ in 30 mM HEPES buffer (pH 7.4)) for 2 hr at room temperature as previously described (*Shirato et al., 2006*). Fixed samples were washed with Milli Q water (Merck Millipore Corporation, Tokyo, Japan) three times, dehydrated in ethanol series, infiltrated with propylene oxide, and embedded in Quetol 812 (Nisshin EM Corporation, Tokyo, Japan).

Resin blocks were sectioned at 80 nm thickness with an ultramicrotome (Leica EM UC7, Leica), contrasted with the EM stainer (Nisshin EM) and lead citrate, and observed with a transmission electron microscope (JEM-1400, JEOL Ltd., Tokyo, Japan).

### Survival analysis

The number of newborns was counted every day for 1 week after birth. The genotypes were analyzed at P0 and P7 for living pups, or at the day of death. Survival was analyzed only for mice that received normal parenting (lactation and heating).

### Measurement of blood glucose, the level of insulin and lipoprotein in plasma, and liver glycogen content

Blood was collected using Heparinized Microhematocrit Capillary Tubes (Thermo Fisher Scientific) and centrifuged at 15,000×g to obtain plasma. Blood glucose was measured using the blood glucose meter Glutest Neo Super (Sanwa Kagaku Kenkyusho Co., Ltd., Aichi, Japan). Plasma insulin levels were measured using the ultra sensitive mouse insulin ELISA Kit (Morinaga Institute of Biological Science, Inc., Kanagawa, Japan). Plasma lipoproteins were analyzed using a lipoprotein profiling service, LipoSEARCH (Skylight Biotech Inc, Akita, Japan). This analysis consists of gel filtration chromatography followed by an on-line enzymatic method for simultaneous quantification of cholesterol, TG, and PC, according to the procedure described by *Usui et al. (2002)* with slight modifications. The nomenclature of the fractions was based on the elution pattern of control plasma that was used for quality check. Glycogen content in liver was analyzed using Glycogen Colorimetric/Fluorometric Assay Kit (BioVision, Inc., Milpitas, CA).

### Lipoprotein precipitation

ApoB-containing lipoproteins were precipitated from pooled plasma (approximately 40 µl) using the LDL/VLDL and HDL Purification Kit (Cell Biolabs, Inc., San Diego, CA) according to the manufacturer's protocol. Only the LDL/VLDL purification method was used. Precipitated samples were used for apoB, TG, cholesterol, and PC measurements.

### Measurement of apoB

ApoB levels in plasma were measured by ELISA. Chicken monoclonal anti-apoB antibody (HUC20, Hiroshima Bio-Medical Co., Ltd., Hiroshima, Japan) was used for capture and rabbit anti-apoB antiserum (Abcam plc., Cambridge, UK) was used for detection. Capture antibody was coated on a Costar 96 well EIA/RIA Easy Wash Clear Flat Bottom High Binding Plate (Corning Incorporated, Corning, NY) overnight at 4°C, and a blocking buffer (1× Reagent Diluent Concentrate 2, R&D Systems, Inc., Minneapolis, MN) was plated to inhibit nonspecific binding for 2 hr at room temperature. All the following steps were performed at room temperature. Diluted samples were then added to the plate and incubated for 2 hr. Next, the detection antibody was incubated for 2 hr, followed by an incubation of horseradish peroxidase-conjugated anti-rabbit IgG antibody (GE Healthcare) for 2 hr. Wells were washed with PBS plus 0.05% Tween 20 for three times after all the incubation steps. TMB (3, 3′, 5, 5′-tetramethylbenzidine, Kirkegaard & Perry Laboratories, Inc., Gaithersburg, MD) substrate was added to the plate, and the reaction was stopped with 2 N sulfuric acid (Nacalai tesque). Absorbance at 450 nm was measured on a plate reader ARVO X3, and absorbance at 544 nm was subtracted to normalize background. A standard curve was prepared with plasma from mice at E18.5–E19.5 for the relative quantification.

### NBD-TG quenching assay

The quenching assay based on a previous report (*Athar et al., 2004*). 180 nmol of PC (eggPC from Sigma–Aldrich, DPPC, POPC, PLPC, PAPC, and PDPC from Avanti) and 5.6 nmol of NBD-TG (Setareh Biotech, LLC, Eugene, OR) were mixed and dried in glass vials under a stream of nitrogen gas. Then, drying was completed in a centrifugal evaporator for 15 min. The dried lipid film was resuspended in 400 µl assay buffer (10 mM Tris-HCl (pH7.4), 150 mM NaCl, and 2 mM EDTA), freeze-thawed five times, and then filtered 21 times through a 50 nm pore size polycarbonate membrane (Nucleopore Track-Etch Membrane, GE Healthcare) using a mini-extruder (Avanti). The solution was centrifuged at

16,100×$g$ (18°C, 10 min), and the supernatant was used as PC/NBD-TG vesicles. Fluorescence of vesicles was measured with a plate reader (ARVO X3) using 485 nm excitation and 535 nm emission wavelengths. Total fluorescence was determined by adding 97 µl of isopropanol to 3 µl of vesicles. Quenching of NBD-TG was calculated as follows: % quenching = (total fluorescence—fluorescence of vesicles)/total fluorescence × 100. We noted that the TG to PC ratio largely affects the quenching efficiency, thus the optimal concentration (which is around 3% TG in our experience) might vary between vials.

## Measurement of MTP activity

Approximately, 1 g of mouse liver was homogenized in 5 ml of ice-cold hypotonic buffer (1 mM Tris-HCl [pH7.4], 1 mM MgCl$_2$ [Wako], and 1 mM EGTA [Sigma–Aldrich]) using a Potter-Elvehjem (Teflon/Glass) tissue grinder. After ultracentrifugation at 218,800×$g$ (10°C, 1 hr), the supernatant was collected and used as an MTP source. Protein concentration was measured using the Bio-Rad Protein Assay. The supernatant of boiled liver homogenate (95°C, 5 min) centrifuged at 16,100×$g$ (18°C, 10 min) was used as a negative control. MTP assay was performed by a modification of previous methods (*Atzel and Wetterau, 1993*, *1994*; *Athar et al., 2004*). MTP activity was measured as the transfer of NBD-TG from donor to acceptor vesicles. PC/NBD-TG vesicles prepared for the quenching assay were used as donor vesicles. Acceptor vesicles were prepared with egg yolk PC (4.385 µmol) dried in glass vials under nitrogen gas stream and then under vacuum for 15 min. The dried PC was rehydrated in 500 µl of sucrose buffer (180 mM sucrose, 10 mM Tris-HCl (pH7.4), 150 mM NaCl, and 2 mM EDTA), freeze-thawed five times, and then filtered 21 times through a 400 nm pore size polycarbonate membrane (Nucleopore Track-Etch Membrane, GE Healthcare) using a mini-extruder (Avanti Polar Lipids). Vesicles were diluted 10-fold in assay buffer (10 mM Tris-HCl (pH7.4), 150 mM NaCl, and 2 mM EDTA), pelleted at 16,100×$g$ (18°C, 10 min), and washed with assay buffer. For each reaction, donor vesicles (containing 2.7 pmol of PC and 0.084 pmol of NBD-TG), acceptor vesicles (containing 43.5 pmol of PC), liver homogenate (100 µg), and 0.05 mg BSA (Sigma–Aldrich) were combined and adjusted to a total volume of 50 µl with assay buffer. Donor vesicle-free reactions were also prepared and used for blanks. The reactions were incubated in a Deep Well Maximizer incubator/mixer (Taitec Co., Ltd., Saitama, Japan) for 0 or 1 hr (60 rpm, 37°C) and then centrifuged at 16,100×$g$ (18°C, 10 min). The supernatants (which contained the donor vesicles) were collected, and the acceptor vesicle-containing pellets were washed with 100 µl of assay buffer and then resuspended and collected in 10 µl of assay buffer. 100 µl of 10% assay buffer/90% isopropanol was applied to wash the tube walls and also collected. The collected supernatant and pellet fractions were diluted 10-fold with isopropanol and total fluorescence of supernatant (donor), pellet (acceptor), and tube wall washes were measured with a plate reader (ARVO X3) using 485 nm excitation and 535 nm emission wavelengths. The fluorescence value of each fraction was corrected for background by subtracting blank well values. NBD-TG transfer was calculated by the differences in fluorescence between 0 and 1 hr incubation. Transfer was calculated as follows: % transfer = fluorescence of pellet fraction/(fluorescence of pellet + fluorescence of supernatant + fluorescence of tube wall wash) × 100.

## Statistical analysis

Unpaired *t*-tests were used to compare two groups. Multiple comparisons were performed with Dunnett's multiple comparison tests, Tukey's multiple comparison tests, or Bonferroni's multiple comparison tests, depending on the combinations of comparisons, after one-way or two-way ANOVA. All analyses were done with GraphPad Prism 5 or 6 for Mac OS X software (GraphPad Software, Inc., La Jolla, CA).

## Acknowledgements

We are grateful for the constructive comments of K Waku, S Narumiya, H Bito, K Ueki, H Okazaki, M Goto, S Takemoto-Kimura, M Matsumoto, T Dohi, and M Sakai. We thank M Mishina for providing Telencephalin-Cre mice, J-I Miyazaki for the expression vectors, J Chun for RH 7777 cells, and F Zhang for depositing Addgene plasmid 48139. We thank K Shimotohno, T Wakita, and H Nishitsuji for advices in cell culture and genome editing, and C Oyama for assistance in histology. We thank F Tokumasu, M Yamada, Y Hishikawa, WJ Valentine, K Yoshida, Y Takahashi,

A Kobayashi, Y Sugimoto, K Nagata, and all members of Shimizu laboratory for advices, discussion, and technical support. This study was supported by JSPS KAKENHI Grant Numbers 24229003 (TS), 26460380 (HS), 26860202 (DH), and 26870879 (TH-Y); National Center for Global Health and Medicine grants 24-001 (TS), 25-201 (TS), 26A105 (TO); the Takeda Science Foundation (TS); CREST, JST (HS).

## Additional information

### Competing interests

RM: Department of Lipidomics, the University of Tokyo is financially supported by Shimadzu Co., and ONO Phamraceutical Col. Ltd. FH: Department of Lipidomics, the University of Tokyo is financially supported by Shimadzu Co., and ONO Phamraceutical Col. Ltd. SMT: Department of Lipidomics, the University of Tokyo is financially supported by Shimadzu Co., and ONO Phamraceutical Col. Ltd. ME: Department of Lipidomics, the University of Tokyo is financially supported by Shimadzu Co., and ONO Phamraceutical Col. Ltd. YK: Department of Lipidomics, the University of Tokyo is financially supported by Shimadzu Co., and ONO Phamraceutical Col. Ltd. TS: Department of Lipidomics, the University of Tokyo is financially supported by Shimadzu Co., and ONO Phamraceutical Col. Ltd. The other authors declare that no competing interests exist.

### Funding

| Funder | Grant reference | Author |
| --- | --- | --- |
| Japan Society for the Promotion of Science (JSPS) | 24229003 | Takao Shimizu |
| Takeda Science Foundation | Special Research grant 2013 | Takao Shimizu |
| Japan Society for the Promotion of Science (JSPS) | 26460380 | Hideo Shindou |
| Japan Society for the Promotion of Science (JSPS) | 26860202 | Daisuke Hishikawa |
| Japan Society for the Promotion of Science (JSPS) | 26870879 | Tomomi Hashidate-Yoshida |
| National Center for Global Health and Medicine | 24-001,25-201 | Takao Shimizu |
| National Center for Global Health and Medicine | 26A105 | Tadashi Okamura |
| Japan Science and Technology Agency (JST) | CREST | Hideo Shindou |

The funders had no role in study design, data collection and interpretation, or the decision to submit the work for publication.

### Author contributions

TH-Y, TH, DH, Conception and design, Acquisition of data, Analysis and interpretation of data, Drafting or revising the article, Contributed unpublished essential data or reagents; RM, FH, SMT, YK, Acquisition of data, Analysis and interpretation of data, Drafting or revising the article, Contributed unpublished essential data or reagents; ME, Acquisition of data, Analysis and interpretation of data, Drafting or revising the article; MT-N, Acquisition of data, Analysis and interpretation of data; RY-T, Performed animal experiments, Acquisition of data; YM, HK, Generated deficient mice, Acquisition of data; TO, Performed animal experiments, Acquisition of data, Analysis and interpretation of data; HS, Conception and design, Drafting or revising the article; TS, Conception and design, Analysis and interpretation of data, Drafting or revising the article

### Ethics

Animal experimentation: All animal experiments were approved by and performed in accordance with the guidelines of the Animal Research Committee of National Center for Global Health and Medicine (12053, 13009, 14045), and the animal experimentation committee of the University of Tokyo (H09-144, P08-042).

# Additional files

**Supplementary file**
• Supplementary file 1. Table of oligo DNA used in this study.

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
