## [Decision Letter]

Thank you for sending your work entitled “LPCAT3 generates arachidonate-enriched phospholipid membranes and regulates triacylglycerol transport” for consideration at *eLife*. Your article has been favorably evaluated by Tadatsugu Taniguchi (Senior editor), a Reviewing editor, and two reviewers.

The following individuals responsible for the peer review of your submission have agreed to reveal their identity: Benjamin Cravatt (Reviewing editor); Charles Serhan (peer reviewer). A further reviewer remains anonymous.

The Reviewing editor and the reviewers discussed their comments before we reached this decision, and the Reviewing editor has assembled the following comments to help you prepare a revised submission.

The reviewers were uniformly in agreement that the manuscript reports significant findings that should be of broad interest to the readership of *eLife*. The conclusions are well-supported by the data. Both reviewers have provided some comments for the authors to consider in revision, which have been inserted below.

1) The role of membrane arachidonate in neurons was previously demonstrated (Mol Biol Cell. 2012 23(24):4689-700.; PLoS One. 2013;8(3):e58425). These papers should be properly referred, and the statement “the functions of membrane arachidonate that are not related to eicosanoid synthesis were unknown” in the Introduction and Discussion should be modified.

2) ∆6-Fatty acid desaturase (FADS2) converts linoleic acid (18:2n-6) to γ-linolenic (18:3n-6), and ∆5-fatty acid desaturase (FADS1) converts dihomo-γ-linolenic acid (20:3n-6) to AA. Both knockout mice cannot synthesize arachidonic acid and thus arachidonic acid-containing phosphatidylcholine. FADS2 knockout mice have been reported by two separate groups mice (49; 50) The knockout mice show reduced content of PUFAs, including AA in tissues, and exhibit a variety of symptoms, including sterility, ulceration, and dermatitis. FADS1 (13) knockout mice began to die gradually starting at 5-6 weeks of age, and show the perturbation in intestinal crypt proliferation and immune cell homeostasis etc. However, either study did not report the phenotype (i.e. accumulation of triacylglycerol in enterocytes). Is really arachidnoic acid-containing phospholipids is important for lipoprotein assembly in the enterocytes? These previous studies should be cited, and the discussion of apparent difference in the enterocyte abnormality between these knockout mice and the LPCAT3 knockout mice should be included. The possibility may exist that continuous remodeling of fatty acid chain is important rather than the amounts of arachidonic acid-containing phospholipids.

3) In Figure 3, the authors showed that LPCAT3 deficiency decreased the levels of 36:5 PC and 38:4 PE, while increased levels of 40:4 PC and 42:5 PC in the proximal small intestine. Which PUFA do these phospholipids contain? MS/MS analyses of these phospholipids should be performed to reveal acyl chain composition. The possibility that other acyltransferase(s) rather than LPCAT3 participates in the formation of 40:4 PC and 42:5 PC should also be discussed.

4) In Figure 4–figure supplement 2, the synthetic pathway of 22:5 is inappropriate. 22:5 is formed by partial β-oxidation of 24:5.

5) In Figure 10, PC with docosahexaenoate promoted TG quenching in liposomes and TG transfer in MTP assay as well as PC with arachidonate. This cannot explain the phenotype of LPCAT3-deficient mice in the proximal small intestine because the level of 22:6 was increased in compensation for the arachidonate decrease in the phospholipid fraction of the proximal small intestine of LPCAT3-deficient mice (Figure 8).

6) In Figure 10, what is the mechanism of the TG transfer-promoting effect of PUFA-containing PC? If this is due to high surface curvature, does the diameter of donor liposomes affect TG transfer efficiency in the MTP assay?

7) In the MTP assay, are similar results observed if heavy donor liposomes and light acceptor liposomes are used?

8) In Figure 3, the authors showed that LPCAT3 deficiency increased LPC level in liver and small intestine. Is there any possibility that increased LPC level affects the clustering or transport of TG?

9) The experiments reported in Figure 4, the authors discover that arachidonate levels are dramatically reduced in the LPCAT3-deficient mice, and importantly reached an increase in n-6 DPA (22:5) and AD (22:4 n-6) these are important observations the authors may choose to elaborate on this point on further in their discussion. Also, the increase in 22:6 n-3 in LPCAT3-deficient mice (Figure 6) suggests that the n-3 derived pro-resolving mediator pathways might be unregulated or enhanced with the production of resolvin and protectins in certain tissues in these mice, which the authors may choose to consider for their discussion. This might be relevant in tissue inflammation and its resolution.

---

## [Author Response]

*1) The role of membrane arachidonate in neurons was previously demonstrated (Mol Biol Cell. 2012 23(24):4689-700.; PLoS One. 2013;8(3):e58425). These papers should be properly referred, and the statement “the functions of membrane arachidonate that are not related to eicosanoid synthesis were unknown” in the Introduction and Discussion should be modified*.

We agreed to the comment, and added the appropriate references and modified the text with a modest and accurate expression in the Introduction.

*2) ∆6-Fatty acid desaturase (FADS2) converts linoleic acid (18:2n-6) to γ-linolenic (18:3n-6), and ∆5-fatty acid desaturase (FADS1) converts dihomo-γ-linolenic acid (20:3n-6) to AA. Both knockout mice cannot synthesize arachidonic acid and thus arachidonic acid-containing phosphatidylcholine. FADS2 knockout mice have been reported by two separate groups mice (*[49]*;*
[50]*) The knockout mice show reduced content of PUFAs, including AA in tissues, and exhibit a variety of symptoms, including sterility, ulceration, and dermatitis. FADS1 (*[13]*) knockout mice began to die gradually starting at 5-6 weeks of age, and show the perturbation in intestinal crypt proliferation and immune cell homeostasis etc. However, either study did not report the phenotype (i.e. accumulation of triacylglycerol in enterocytes). Is really arachidnoic acid-containing phospholipids is important for lipoprotein assembly in the enterocytes? These previous studies should be cited, and the discussion of apparent difference in the enterocyte abnormality between these knockout mice and the LPCAT3 knockout mice should be included. The possibility may exist that continuous remodeling of fatty acid chain is important rather than the amounts of arachidonic acid-containing phospholipids*.

As suggested by the reviewer, it would be interesting to compare the phenotypes of LPCAT3-deficient mice and FADS-deficient mice. It is important that in FADS-deficient mice, PUFA are provided from the breast milk, and the deficiency occurs only after weaning. Thus, the neonatal intestinal abnormalities might not be seen in FADS-deficient mice naturally. In addition, as intestinal lipid accumulation and enterocyte dysfunction occur only when high amounts of lipids are absorbed, it is possible that a high-fat diet might cause such abnormalities in FADS-deficient mice. In fact, lipid droplet accumulation in the liver has been reported in FADS2-deficient mice (Stoffel, 2008), which was rescued by arachidonate feeding. This observation appears consistent with our model. We also agree with the idea that continuous remodeling is important, as will be described below. These points are appropriately added in the Discussion.

*3) In*
Figure 3*, the authors showed that LPCAT3 deficiency decreased the levels of 36:5 PC and 38:4 PE, while increased levels of 40:4 PC and 42:5 PC in the proximal small intestine. Which PUFA do these phospholipids contain? MS/MS analyses of these phospholipids should be performed to reveal acyl chain composition. The possibility that other acyltransferase(s) rather than LPCAT3 participates in the formation of 40:4 PC and 42:5 PC should also be discussed*.

Since MS/MS analysis did not give sufficient signals for very minor species, we alternatively performed SRM analysis for selected PC and PE molecules, with selection of fatty acid fragments at Q3. In addition to those pointed out by the reviewer, we analyzed multiple PC and PE species with changed levels to be more complete (Figure 4 and Tables 1 and 2). This revealed that these species are a combination of various isobaric species. Interestingly, this analysis revealed that the changes have different trends depending on the fatty acid composition, with phospholipids containing C20 PUFAs decreased, and those containing C22 or C24 PUFAs increased.

In answer to the second point, we show that in liver neutral lipid fractions (which reflect fatty acids incorporated during de novo synthesis, Figure 7), the C22 PUFAs are increased. This strongly suggests that the increased C22 PUFAs were used by LPAATs during de novo synthesis, which explains the increased species such as 40:4 PC and 42:5 PC. We cannot exclude the possibility that other unknown LPCATs in the remodeling pathway have also incorporated these C22 PUFAs in PC and PE, although this sounds unlikely since most LPCAT and LPEAT activity is absent in LPCAT3-deficient mice. The SRM data is presented in Tables 1 and 2, and an explanation of the formation of species such as 40:4 PC is shown in Figure 12 in the revised manuscript.

*4) In Figure 4–figure supplement 2, the synthetic pathway of 22:5 is inappropriate. 22:5 is formed by partial β-oxidation of 24:5*.

We thank the reviewer for pointing out our mistake, which was adjusted in the revised manuscript. We moved the figure to Figure 3—figure supplement 2, since it appears earlier, related to the new table in response to comment 3.

*5) In*
Figure 10*, PC with docosahexaenoate promoted TG quenching in liposomes and TG transfer in MTP assay as well as PC with arachidonate. This cannot explain the phenotype of LPCAT3-deficient mice in the proximal small intestine because the level of 22:6 was increased in compensation for the arachidonate decrease in the phospholipid fraction of the proximal small intestine of LPCAT3-deficient mice (*Figure 8*)*.

This is an important point of this study. This observation led us to propose the model of the “local accumulation” of PUFA-containing phospholipids caused by the continuous remodeling by LPCAT3. Since this concept might be difficult for broad readers to understand in plain text, we added a figure explaining this (Figure 13). We think that 22:6 (and also the other C22 PUFAs increased in LPCAT3-deficient mice) cannot compensate arachidonate, since they cannot accumulate locally and do not reach a sufficient concentration to promote TG transfer in lipoproteins.

*6) In*
Figure 10*, what is the mechanism of the TG transfer-promoting effect of PUFA-containing PC? If this is due to high surface curvature, does the diameter of donor liposomes affect TG transfer efficiency in the MTP assay*?

This issue remains unclear and needs to be clarified in the future studies. Molecular dynamics simulations showed that local clustering promotes flip-flop of TG (Khandelia et al.). Therefore, we speculate that PUFA-containing PC promotes the clustering of TG (as shown by the quenching assay) and that the high flip-flop rate enables efficient TG transfer since MTP should have access for TG only in the luminal leaflet. Therefore the surface curvature might not necessarily be involved in transfer by, but rather in TG clustering. Indeed, although we analyzed TG quenching and transfer using liposomes of different diameters using different pore size filters (30-100 nm), we found no difference with statistical significance. However, we know empirically that pressure during liposome extrusion might affect NBD-TG quenching and that extrusion pressure changes depending on the filter pore size. Therefore, two factors (diameter and pressure) are changed in the experimental design, making it difficult to draw a precise conclusion, and we prefer not to include this data in the manuscript. In addition, it is also difficult to know how liposome diameter affects local curvature on the surface of clustered TG.

*7) In the MTP assay, are similar results observed if heavy donor liposomes and light acceptor liposomes are used*?

According to the suggestion by the reviewer, we tried the experiment, but found problems in the experimental design. To enable heavy liposomes to be pelleted down during centrifugation, we needed a filter size of 400 nm in minimum. It is known that under these conditions, liposomes are multilamellar, which are inadequate as donors since only the outermost layer can provide TG. As expected, we could not observe TG transfer when using these multilamellar heavy donors. Unfortunately, this made it impossible to compare heavy donors with different PC composition.

*8) In*
Figure 3*, the authors showed that LPCAT3 deficiency increased LPC level in liver and small intestine. Is there any possibility that increased LPC level affects the clustering or transport of TG*?

According to the reviewer’s suggestion, we analyzed the effect of LPC on TG clustering and transport. We could not observe any difference when using up to 5% of LPC mixed with egg PC. We described it in the text under the subsection headed “Membrane PUFAs enable efficient clustering and transport of TGs”.

*9) The experiments reported in*
Figure 4*, the authors discover that arachidonate levels are dramatically reduced in the LPCAT3-deficient mice, and importantly reached an increase in n-6 DPA (22:5) and AD (22:4 n-6) these are important observations the authors may choose to elaborate on this point on further in their discussion. Also, the increase in 22:6 n-3 in LPCAT3-deficient mice (*Figure 6*) suggests that the n-3 derived pro-resolving mediator pathways might be unregulated or enhanced with the production of resolvin and protectins in certain tissues in these mice, which the authors may choose to consider for their discussion. This might be relevant in tissue inflammation and its resolution*.

We added discussion about how these increased C22 PUFAs are incorporated into phospholipids. In addition, we added discussion about n-3 derived pro-resolving mediators and 18:2-derived lipid mediators. Since it is important that in contrast to the decreases in arachidonate, the changes in 18:2 and 22:6 are seen only in a subset of tissues, we added figures of GC-FID data showing the changes in these fatty acids to avoid misleading of the readers (Figure 5).